# Occurrence and repair of alkylating stress in the intracellular pathogen *Brucella abortus*

Katy Poncin[1,2], Agnès Roba[1], Ravikumar Jimmidi[3], Georges Potemberg[1], Antonella Fioravanti [4,5], Nayla Francis[1], Kévin Willemart[1], Nicolas Zeippen[1], Arnaud Machelart [1,6], Emanuele G. Biondi[4], Eric Muraille[7,8], Stéphane P. Vincent[2] & Xavier De Bolle [1]*

It is assumed that intracellular pathogenic bacteria have to cope with DNA alkylating stress within host cells. Here we use single-cell reporter systems to show that the pathogen *Brucella abortus* does encounter alkylating stress during the first hours of macrophage infection. Genes encoding direct repair and base-excision repair pathways are required by *B. abortus* to face this stress in vitro and in a mouse infection model. Among these genes, *ogt* is found to be under the control of the conserved cell-cycle transcription factor GcrA. Our results highlight that the control of DNA repair in *B. abortus* displays distinct features that are not present in model organisms such as *Escherichia coli*.

[1] URBM, Narilis, University of Namur, Namur, Belgium. [2] Sir William Dunn School of Pathology, University of Oxford, South Parks Road, Oxford OX1 3RE, UK. [3] Unité de Chimie Organique, University of Namur, 61 rue de Bruxelles, 5000 Namur, Belgium. [4] Unité de Glycobiologie Structurale et Fonctionnelle, UMR 8576 CNRS, Université de Lille, 50 Avenue Halley, Villeneuve d'Ascq, France. [5] VIB, Vrije Universiteit Brussel, Pleinlaan 2, 1050 Brussels, Belgium. [6] Université de Lille, CNRS, INSERM, CHU Lille, Institut Pasteur de Lille, U1019, UMR 8204, Center for Infection and Immunity of Lille, Lille, France. [7] IMM, 31 Chemin Joseph Aiguier, 13009 Marseille, Aix-Marseille Université, Marseille, France. [8] Laboratoire de Parasitologie, Faculté de Médecine, Université Libre de Bruxelles, Brussels, Belgium. *email: xavier.debolle@unamur.be

On DNA, alkylating stress typically results in aberrant methylation patterns. Many positions of DNA can be targeted and these modifications range from innocuous to mutagenic or cytotoxic[1,2]. Alkylating agents are ubiquitous in the environment and also have endogenous and dietary sources[3,4].

Three main endogenous sources of alkylating agents are reported in living organisms: (1) the ubiquitous methyl-donor S-adenosylmethionine (SAM)[5], (2) lipid peroxidation-derived alkylating agents[6], and (3) N-nitroso compounds resulting from the nitrosation of metabolites and being either direct alkylating agents or requiring metabolic activation[7–9]. SAM is often cited as the primary cause of endogenous alkylation, but its role has probably been overestimated in prokaryotic cells. Indeed, a 100-fold change of SAM levels was achieved experimentally without causing any change in the mutation rate of *Escherichia coli*[10]. Additionally, lipid peroxidation predominantly occurs on poly-unsaturated fatty acids[11], which are rarely present in bacteria[12,13]. Prokaryotic cells are thus considered to face marginal lipid-derived alkylating stress. In fact, in *E. coli*, the majority of the spontaneous mutations resulting from alkylating stress was found to be generated via the endogenous formation of N-nitroso compounds[7,14].

One question that remains unanswered is whether alkylating agents are produced by immune cells to fight intracellular pathogens. Of relevance, macrophages and neutrophils are known to produce N-nitroso compounds[15,16]. However, the occurrence of alkylating stress on intracellular bacteria has not been detected as yet. It has been hypothesized that N-nitroso compounds could be differentially produced in subcellular compartments[17], suggesting that vacuoles containing bacteria may also be prone to accumulating such compounds. Indeed, many intracellular bacteria first travel through an endosomal-derived vacuole, before reaching their replicative compartment[18], as in *Brucella abortus*, the causative agent of brucellosis in animals and Malta fever in humans[19,20]. This class III α-proteobacterium is known to enter host cells to form an endosomal *Brucella*-containing vacuole (eBCV), which becomes acidified to a pH of 4.0–4.5, suggesting that the eBCV undergoes normal endosomal maturation process, with the acquisition of the LampI marker[21,22]. Later, in most cell types, the bacterium reaches the endoplasmic reticulum, its replicative niche (rBCV)[19,23,24].

Both prokaryotic cells and eukaryotic cells are competent for N-nitrosation reactions. In eukaryotic cells, the production of N-nitroso compounds is dependent on an acidic pH and the intracellular concentration of reactive nitrogen species (RNS)[15,25]. However, in *E. coli*, a side reaction of nitrate reductases leads to the N-nitrosation of amines at a neutral pH[7]. Three classes of *E. coli* mutants have been shown to be deficient in nitrosation: (1) *narG*, encoding the catalytic subunit of nitrate reductase A; (2) *fnr*, encoding a pleotropic activator that influences the expression of the *narGHIJ* operon; and (3) *moa*, the mutant with the strongest nitrosation deficiency, probably because the enzyme is involved in the synthesis of a molybdopterin cofactor, required by all three *E. coli* nitrate reductases[7]. Importantly, the endogenous production of N-nitroso compounds is known to be more important in anaerobic and resting *E. coli*[26]. In this respect, during the first hours of infection, while still inside the eBCV, *B. abortus* is known to be blocked in the G1 stage of its cell cycle, characterized by a non-growing and non-replicating state[27].

In the late 1970s, a specific response to alkylating stress was described in vitro for *E. coli*[28]. This system was called "adaptive response" and is based on the detection of a methylphosphotriester (meP3ester) modification on DNA by the Ada[29]. Most bacteria possess an Ada-based adaptive response, with some variation in the target genes and their genomic organization[1]. In *E. coli*, *ada* is stochastically expressed to produce on average only one Ada protein per generation[30]. The detected meP3ester group is captured on the cysteine 38 (C38) residue of Ada, which becomes active as a transcription factor, upregulating the expression of a series of genes coding for proteins dedicated to the repair of alkylated DNA. These proteins comprise Ada itself, which can also directly repair $O^6$-methylguanine and $O^4$-methylthymine via the capture of the methyl group by the C321 residue of Ada; the dioxygenase AlkB, involved in the direct repair of the mutagenic lesions $N^1$-methyladenine and $N^3$-methylcytosine, as well as the DNA glycosylase AlkA, which removes the most cytotoxic lesion $N^3$-methyladenine via the base excision repair (BER) pathway[31]. The *aidB* gene is also over-expressed by the adaptive response, but its function still remains elusive[32,33]. There are also two proteins constitutively produced and independent of the adaptive system, namely the methyl-transferase Ogt, which has a C139 residue with similar function than the C321 of Ada, and the glycosylase TagA, which is functionally similar to AlkA[31]. Importantly, other DNA repair pathways can be involved in repairing alkylated DNA. For instance, in *E. coli*, the SOS pathway—which is repressed by LexA under non-stress conditions—is activated early following alkylating stress, before the adaptive response takes over[34].

Here, we demonstrate that a weak alkylating stress occurs when *B. abortus* is inside its eBCV in a macrophage cell line. We also show that genes responsible for the response against alkylating stress are required by *B. abortus* to survive following mice intranasal infection. Our data indicate that *B. abortus* does not possess a functional Ada-based adaptive system, but instead relies on redundant repair pathways, partially dependent on the methylation-sensitive transcription factor GcrA and the SOS response, to cope with alkylating stress and subvert potentially mutagenic host environments.

## Results

**Conservation of alkylated DNA repair genes in bacteria**. We rationalized that if most intracellular bacteria face alkylating stress, there would be a significant conservation of some alkylated DNA repair genes. We found that many intracellular bacteria, including obligate pathogens such as *Chlamydia pneumoniae* and *Coxiella burnettii*, have genes homologous to some of known DNA repair genes (Fig. 1). Note that *Brucella* species are predicted to be particularly well equipped against alkylating stress (Fig. 1 and Supplementary Fig. 1).

**Alkylation stress is encountered by *Brucella* inside host cells**. Until now, strategies to detect the presence of alkylating stress inside host cells have been based on the survival of alkylation-specific DNA repair enzymes. These approaches have been uninformative, probably because repair systems are redundant, or because the stress is too weak to be detected by CFU counting[35,36]. Here, we took advantage of the ability of the auto-regulated Ada protein from *E. coli* to detect meP3ester groups on DNA[37] to create a transcription-based fluorescent reporter system. As the *ada* gene is in operon with *alkB* in *E. coli*[38], we replaced *alkB* with a superfolder *gfp* on a medium-copy plasmid (Fig. 2a). A mutated version of the reporter system was used as a negative control, in which a C38A mutation was introduced in Ada to prevent the protein from capturing meP3ester groups.

The reporter system was first tested with the alkylating agent methyl methane sulfonate (MMS) in *E. coli* and in *Salmonella enterica* biovar Typhimurium, which does not possess an Ada-based functional adaptive system[39]. In both bacteria, the reporter system was activated only in the presence of MMS and only with the functional version of *E. coli* Ada (Supplementary Fig. 2). In *B.*

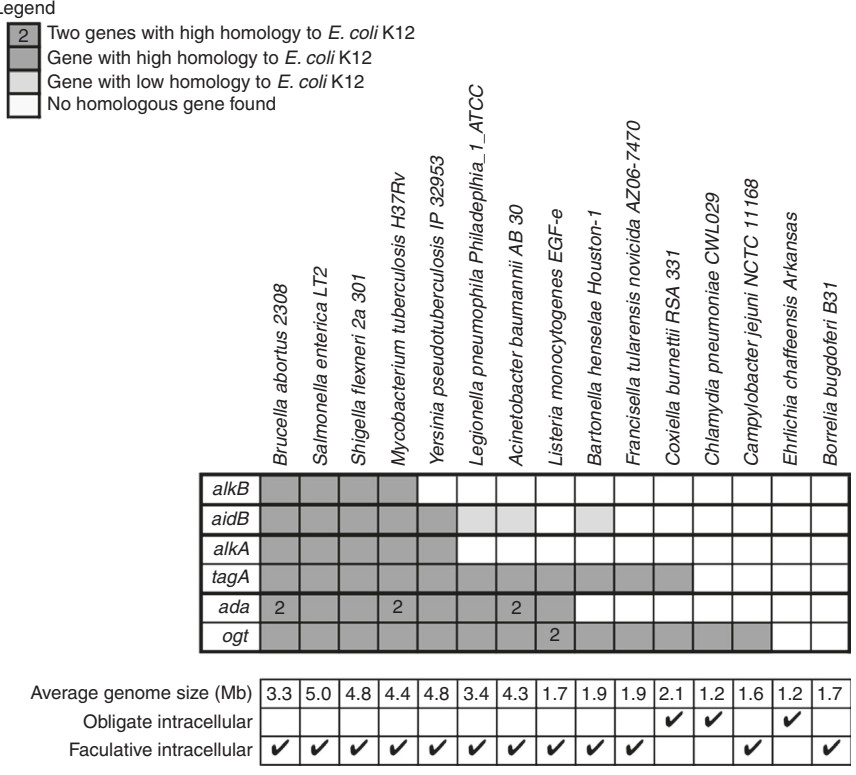

**Fig. 1** Conservation of genes coding for alkylated DNA repair proteins. Genes were grouped by function. Homology was calculated based on *E. coli* K12 genome ([www.patricbrc.org/](www.patricbrc.org/)). In the case of *aidB*, genes annotated as acyl-coA dehydrogenase with *e*-value between $10^{-29}$ and $10^{-44}$ were considered as genes with low homology and genes with *e*-value lower than $10^{-133}$ were considered as genes with high homology

*abortus*, the reporter was more active in exponential phase cultures compared to stationary phase cultures (Supplementary Fig. 3a). The emitted fluorescence was also dependent on the time of exposure and concentration of MMS (Supplementary Fig. 3b). To check that the reporter system was not affected by the endogenous Ada production in *B. abortus*, the mean fluorescence intensity of the system in a *B. abortus* Δ*ada1* Δ*ada2* background was compared with results in a wild-type (WT) background. No statistical difference could be observed between the two experiments, supporting the notion that endogenous Ada proteins do not affect the activation of the reporter system (Supplementary Fig. 3c).

The reporter system was then tested at the single-cell level during infection of RAW 264.7 macrophages. The first time point was 5 h PI, which corresponds to the end of the first phase of the infection, when *B. abortus* is not growing and blocked in a G1-like phase inside the eBCV[27]. The second time point chosen was at 24 h PI, when the bacteria are actively growing inside the rBCV[27]. The ratio between the mean fluorescence intensities of the functional reporter system and the non-functional version was calculated, and the level of fluorescence was significantly ($p <$ 0.01) higher at 5 h PI than at 24 h PI (Fig. 2b, Supplementary Fig. 3d). This suggests that bacteria encounter alkylating stress inside host cells, but mainly during the initial phase of the infection.

To investigate the potential mutagenic properties of the intracellular eBCV environment compared to other conditions, we sequenced the genome of several individual clones of *B. abortus* before and after infection of RAW 264.7 macrophages at 6 and 48 h, and after early infection of mice (60 h), as well as after a similar number of generations in liquid culture (48 h). The genomes of five individual clones resulting from these different conditions were sequenced and subsequent analyses indicated that the number of mutations was not increased in the infection conditions compared to the culture (Supplementary Fig. 4).

**N-nitrosation events occur inside the eBCV.** One of the main sources of alkylating agents is the N-nitrosation of metabolites[8]. Since the content of the eBCV is unknown, we developed a new tool to investigate the presence of N-nitrosation in this compartment. Succinimidyl ester groups have been successfully employed to label the outer membrane of bacteria with fluorescent molecules[27,40]. Besides, Miao et al.[41] established a highly specific probe emitting fluorescence upon N-nitrosation. The two techniques were combined to create a N-nitrosation-sensitive probe that was covalently attached to the surface of *B. abortus*, allowing us to follow whether N-nitrosation occurs inside the eBCV, before the growth of bacteria[27] (Fig. 3a).

Labeled bacteria were first tested for their fluorescence in presence of $KNO_2$, which generates the NO donor $N_2O_3$ in aqueous solution (Supplementary Fig. 5a). Autofluorescence of non-labeled *B. abortus* was also compared to the fluorescence of labeled bacteria in the absence of $KNO_2$ (Supplementary Fig. 5a). Results demonstrate that the probe emits fluorescence when N-nitrosated, as expected. Next, RAW 264.7 macrophages were infected with labeled bacteria and mean fluorescence intensities were calculated at 5 h post infection at the single-cell level (Fig. 3b, Supplementary Fig. 5b, c). As a negative control, the same experiment was conducted in the presence of 163 μM of ascorbate in the cell culture medium, as this concentration of antioxidant is known to inhibit N-nitrosation reactions in RAW 264.7 macrophages[42]. We observed that about a quarter (23.3 %) of the bacterial population was subjected to N-nitrosation inside

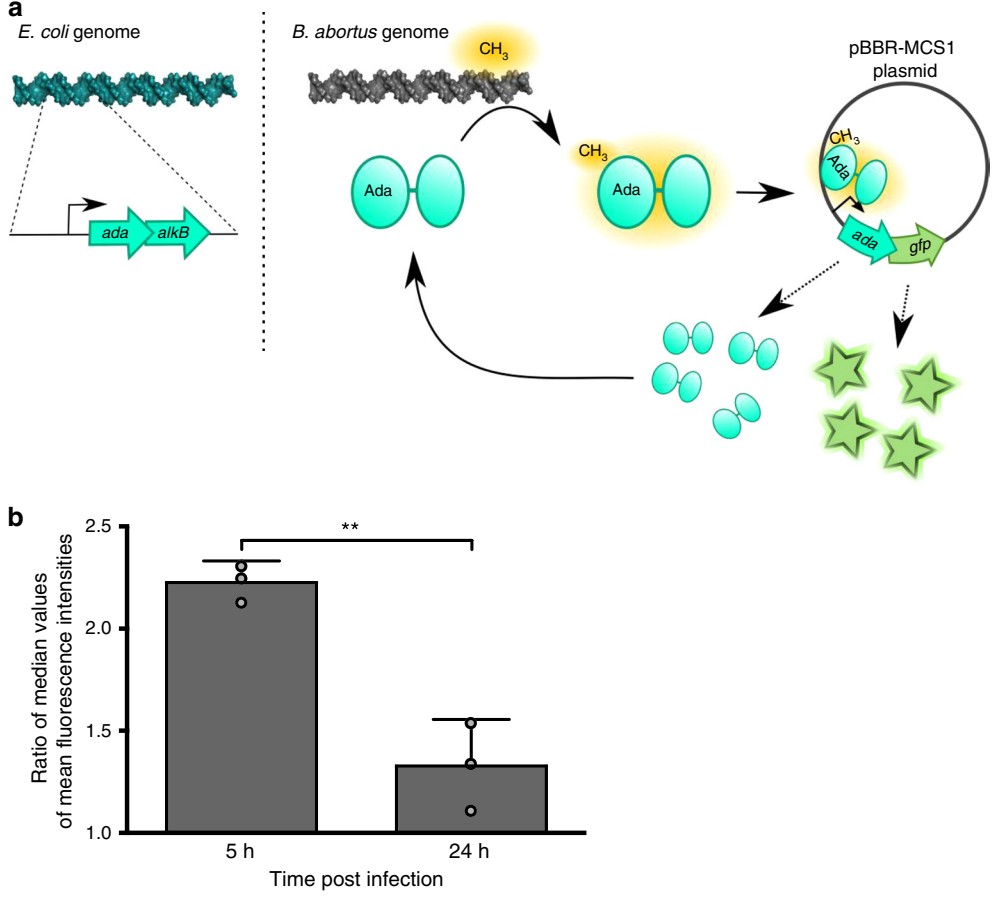

**Fig. 2** Reporter system for alkylation stress. **a** Schematic representation of the reporter system. The sequence corresponding to $ada_{E.\ coli}$ and its promoter were cloned into a pBBR-MCS1 plasmid and a superfolder *gfp* was inserted downstream $ada_{E.\ coli}$. This plasmid (pBBR-$p_{ada}$-*ada*-*gfp*) was transferred to *B. abortus*. When $Ada_{E.\ coli}$ detects a methylphosphotriester group on *B. abortus* DNA, it activates the expression of its own promoter, which leads to an accumulation of $Ada_{E.\ coli}$ and GFP. Note that a mutation in $ada_{E.\ coli}$ (C38A) leads to the abrogation of its ability to bind methylphosphotriester. **b** Bacteria carrying either the pBBR-$p_{ada}$-*ada*-*gfp* reporter system or its mutated version ($ada^{C38A}$) were used to infect RAW 264.7 macrophages and mean fluorescence intensities (FITC channel) were calculated at 5 or 24 h post infection ($n = 60$). Ratio of median values (*ada*/*ada*$^{C38A}$) were plotted for biological triplicates. Error bars correspond to standard deviation. Student's *t*-test was performed with $p < 0.01$ (\*\*). Source data are provided as a Source Data file

the eBCV. Importantly, the addition of ascorbate markedly decreased the proportion of positive labeled bacteria (Fig. 3b, Supplementary Fig. 5c), indicating that N-nitrosation reactions can effectively be prevented by antioxidants. LampI labeling of BCVs confirmed that most *B. abortus* were in the endosomal stage (eBCV) of the infection at that time point (Supplementary Fig. 5d).

We also investigated the presence of alkylating stress due to the endogenous production of N-nitroso compounds. To do so, the $Ada_{E.\ coli}$-based reporter system was tested in different genetic backgrounds (WT, Δ*narG*, and Δ*moaA*) at 5 h post infection. The deletion of *narG* alone was not sufficient to reduce alkylating stress, whereas it was the case with the Δ*moaA* strain (Fig. 3c). Notably, the addition of 163 μM of ascorbate to the cell culture medium did not significantly reduce the extent of alkylating stress, suggesting that external N-nitrosation is not responsible for alkylating stress in these conditions. Overall, this indicates that during RAW 264.7 infection, alkylating stress is mainly produced endogenously by *B. abortus* metabolism.

**Key actors against alkylating stress in *B. abortus*.** To evaluate which DNA repair genes are required by *B. abortus* to counteract alkylating stress, deletion strains were constructed and plated on

rich medium supplemented with alkylating agents (Fig. 4). Mutants were constructed for genes predicted to encode proteins involved in direct repair, BER, HR, nucleotide excision repair (NER), and mismatch repair (MMR). Two strains were also included as negative controls: the triple mutant Δ*mutM* Δ*mutY* Δ*mutT*, required for DNA repair following oxidative stress, and the Δ*virB* strain, lacking the *B. abortus* type IV secretion system. These strains were tested for their survival against the $S_{N}1$ agent methylnitronitrosoguanidine (MNNG) that reacts with DNA in two main steps via a unimolecular nucleophilic substitution, and the $S_{N}2$ agent MMS, which reacts in one step with biomolecules[43,44] (Fig. 4).

Interestingly, some genes were required against MMS only, such as *alkB* and the BER genes. A *B. abortus* Δ*xthA*1 endonuclease mutant had been previously reported to be sensitive to MMS[45], although the function of XthA2 remains unclear. Here, we show that XthA1 is the major endonuclease, since its deletion was sufficient to confer sensitivity to MMS, whereas it was not the case for the deletion of *xthA*2. However, the double mutant was approximately a hundred-fold more sensitive than the single Δ*xthA*1 (Fig. 4), indicating that the genes have partially redundant functions. Similarly, the Δ*alkA* mutant was not affected by MMS, while the Δ*tagA* mutant displayed a 35-fold decrease in bacterial survival recovery. The Δ*tagA* Δ*alkA* mutant

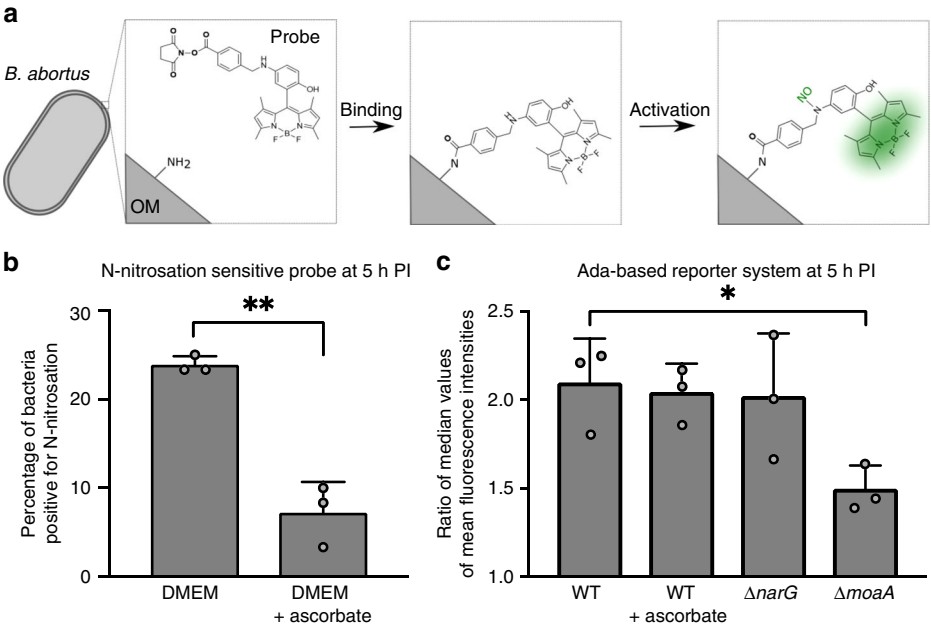

**Fig. 3** Production of N-nitroso compounds inside host cells. **a** Schematic representation of the N-nitrosation-sensitive probe reacting with primary amine from *B. abortus* outer membrane (OM), and subsequently being activated by NO. **b** Evaluation of exogenous N-nitrosation by calculating the percentage of positive labeled bacteria. Bacteria were labeled with the N-nitrosation-sensitive probe and used to infect RAW 264.7 macrophages. Mean fluorescence intensities were calculated at 5 h post infection ($n = 60$) and values above 100 were considered as positive. The addition of 163 µM of ascorbate to the cell culture medium at time 0 was used to inhibit N-nitrosation. Experiments were done in biological triplicates. Error bars correspond to standard deviation. Student's *t*-test was performed with $p < 0.01$ (**). Source data are provided as a Source Data file. **c** Evaluation of endogenous N-nitroso compounds formation via the alkylation-sensitive reporter system. The Ada$_{E. coli}$-based reporter system was used in three genetic backgrounds (WT, ΔnarG, and ΔmoaA) and in the presence of ascorbate (for the WT background only). Bacteria carrying either the pBBR-p$_{ada}$-ada-gfp reporter system or its mutated version (ada$^{C38A}$) were used to infect RAW 264.7 macrophages and mean fluorescence intensities were calculated at 5 h post infection ($n = 60$). Ratio of median values (ada/ada$^{C38A}$) was plotted for biological triplicates. Error bars correspond to standard deviation. A Student's *t*-test was performed with a $p < 0.05$ (*). Source data are provided as a Source Data file

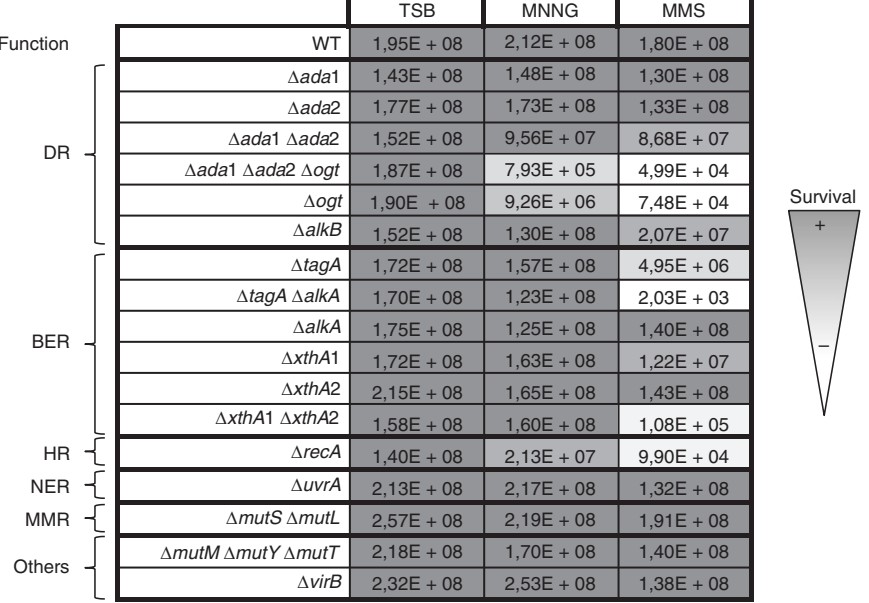

| Function | | TSB | MNNG | MMS |
|---|---|---|---|---|
| | WT | 1,95E + 08 | 2,12E + 08 | 1,80E + 08 |
| DR | Δada1 | 1,43E + 08 | 1,48E + 08 | 1,30E + 08 |
| | Δada2 | 1,77E + 08 | 1,73E + 08 | 1,33E + 08 |
| | Δada1 Δada2 | 1,52E + 08 | 9,56E + 07 | 8,68E + 07 |
| | Δada1 Δada2 Δogt | 1,87E + 08 | 7,93E + 05 | 4,99E + 04 |
| | Δogt | 1,90E + 08 | 9,26E + 06 | 7,48E + 04 |
| | ΔalkB | 1,52E + 08 | 1,30E + 08 | 2,07E + 07 |
| BER | ΔtagA | 1,72E + 08 | 1,57E + 08 | 4,95E + 06 |
| | ΔtagA ΔalkA | 1,70E + 08 | 1,23E + 08 | 2,03E + 03 |
| | ΔalkA | 1,75E + 08 | 1,25E + 08 | 1,40E + 08 |
| | ΔxthA1 | 1,72E + 08 | 1,63E + 08 | 1,22E + 07 |
| | ΔxthA2 | 2,15E + 08 | 1,65E + 08 | 1,43E + 08 |
| | ΔxthA1 ΔxthA2 | 1,58E + 08 | 1,60E + 08 | 1,08E + 05 |
| HR | ΔrecA | 1,40E + 08 | 2,13E + 07 | 9,90E + 04 |
| NER | ΔuvrA | 2,13E + 08 | 2,17E + 08 | 1,32E + 08 |
| MMR | ΔmutS ΔmutL | 2,57E + 08 | 2,19E + 08 | 1,91E + 08 |
| Others | ΔmutM ΔmutY ΔmutT | 2,18E + 08 | 1,70E + 08 | 1,40E + 08 |
| | ΔvirB | 2,32E + 08 | 2,53E + 08 | 1,38E + 08 |

Survival
+
−

**Fig. 4** Survival of DNA repair mutants against alkylating agents in vitro. Deletion strains were plated on rich medium (TSB) supplemented or not with alkylating agents (35 µM of MNNG or 2.5 mM of MMS). Data shown here are the mean values of colony forming units for biological triplicates. DR stands for direct repair, BER for base excision repair, HR for homologous recombination, NER for nucleotide excision repair, and MMR for mismatch repair. The category "others" comprises 8-oxo-dG repair (*mutM mutY mutT*) and the type IV secretion system (*virB*) as negative controls. Source data are provided as a Source Data file

was more markedly sensitive, with a decrease in CFU of 5 orders of magnitude compared to the control condition (Fig. 4). This result indicates that AlkA and TagA share an overlapping function, which is crucial for survival in the presence of MMS in *B. abortus*, as it is the case in *E. coli*[46]. The Δ*recA* mutant was also strongly affected by MMS (as described previously[47]), but only slightly in the presence of MNNG (Fig. 4). The Δ*ogt* mutant is particularly noteworthy, as it appeared to be very sensitive to MMS exposure. The triple mutant Δ*ada*1 Δ*ada*2 Δ*ogt* was only marginally more attenuated than the single Δ*ogt* mutant against MMS, and slightly more against MNNG (Fig. 4). This indicates that the presence of Ogt is a key factor for *B. abortus* survival against alkylating agents in these conditions, unexpectedly more than the two Ada proteins compared to the *E. coli* model. At the protein level, Ogt$_{B. abortus}$ is predicted to be 33% identical to Ogt$_{E. coli}$ with the conservation of the methyl-acceptor C139 residue (Supplementary Fig. 6a). In *B. abortus*, the residue corresponding to Ogt$_{E. coli}$ S134 is a proline (Supplementary Fig. 6a). Remarkably, in *E. coli*, the mutation of S134 into a proline confers broader substrate specificity to the protein by increasing the size of its active site[48], so this could explain why Ogt seems to be the major actor amongst the three methyltransferases of *B. abortus*.

The two genes predicted to code for Ada also possess conserved C38 and C321 residues (Supplementary Fig. 6b). Nevertheless, the deletion of the *ada1* and *ada2* genes did not change drastically the sensitivity of *B. abortus* to alkylating agents (Fig. 4), Questioning whether the Ada proteins are functioning as transcription factors. Quantitative reverse transcription polymerase chain reaction (RT-qPCR) experiments were performed on *B. abortus* in the presence or absence of MMS, and several DNA repair genes were compared for their mRNA levels in these conditions (Fig. 5). Interestingly, the mRNA levels of the two *ada* genes were not statistically increased after MMS exposure. In fact, the only overexpressed alkylation-specific genes were *alkA* and *tagA*, which are predicted to code for proteins of similar function. The absence of induction of the *ada* genes (Fig. 5) and their marginal role in coping with alkylating stress in vitro (Fig. 4) suggest that *B. abortus* does not rely on a classical Ada-dependent adaptive system to subvert alkylating stress. It has been proposed that the absence of an adaptive response in *S. enterica* serovar Typhimurium could be due to the lack of an acidic residue in position 106th[39]. Similarly, in *B. abortus*, the corresponding position is occupied by either a N116 (Ada1) or a V105 (Ada2) (Supplementary Fig. 6b), so this could also explain why *B. abortus* does not possess a functional Ada-based adaptive system.

Another gene that was overexpressed upon MMS exposure was *lexA* (Fig. 5). Interestingly, the early accumulation of the SOS repressor LexA is a marker of the activation of the SOS response, because *lexA* itself is part of its early regulon to prevent an uncontrolled over-activation of the SOS response[49]. To determine which genes are part of the SOS regulon in *B. abortus* under alkylating stress, the fold induction of several genes was compared after MMS exposure in the WT and *lexA* over-expression (pBBRMCS1-$p_{lac}$-*lexA*) strains. In the overexpression strain, *lexA* itself was not further induced by MMS exposure, as its level of transcription was probably already maximal (Supplementary Fig. 7). Importantly, neither *tagA* nor *alkA* were significantly differentialy induced by MMS in the two conditions (Supplementary Fig. 7), indicating that their induction is dependent of a yet unknown factor. Among the two error-prone DNA polymerases of *B. abortus*, the *imuABC* operon[50] had a drop of induction upon *lexA* overexpression, in contrast to *dinB*, which encodes DNA polymerase IV (Supplementary Fig. 7). Of note, in α-proteobacteria, a few genes are downregulated following the activation of the SOS response[51,52]. In *B. abortus*,

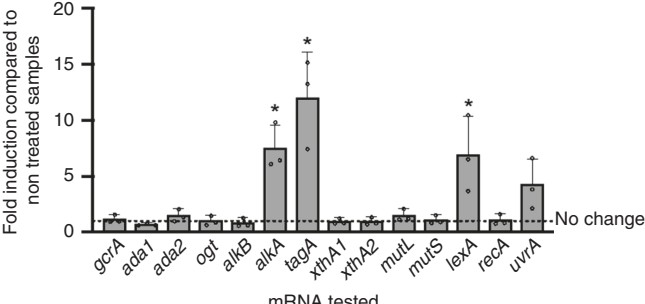

**Fig. 5** Gene expression following MMS treatment. RT-qPCR was performed on exponential phase *B. abortus* cultured in rich medium for 5 h in the presence or absence of 2.5 mM MMS. Experiments were performed three times and mean values were compared between the stressed and non-stressed conditions. Error bars represent standard deviation. Student's *t*-test was performed on data with minimum 1.5-fold induction ($p < 0.05$, *). Source data are provided as a Source Data file

our results suggest that it could also be the case for the NER endonuclease *uvrA*, as it was further induced in the *lexA*-overexpressing strain (Supplementary Fig. 7). Importantly, RecA is suspected to constitutively trigger a basal SOS response in *B. abortus*, even under non-stress conditions[53], which could explain the relatively low induction of many potential target genes in our experiment.

**Involvement of GcrA against alkylation stress in vitro.** One striking characteristic of *ogt* is the presence, right after the start codon, of a GANTC motif. This sequence is known to be a site of epigenetic regulation in α-proteobacteria[54]. Indeed, GANTC sites have been shown to be methylated by CcrM in a cell-cycle-dependent manner in the α-proteobacterium *Caulobacter crescentus*[55,56], and probably also in *B. abortus*[57,58]. Interestingly, the gene coding for the alkylation-specific DNA repair AlkB protein is regulated throughout the cell cycle in *C. crescentus*[59] and in this bacterium, *alkB* mRNA levels drop by more than two-fold in a GcrA depleted strain[60]. Since the cell cycle-dependent transcription factor GcrA is known to be modulated by methylated GANTC sites on *C. crescentus* DNA[60,61], we identified the ortholog of *gcrA* in *B. abortus* (BAB1_0329), a gene previously shown to be essential[62]. ChIP-seq analysis of GcrA was performed to identify its targets in *B. abortus*. As many as 232 hits were found for the first chromosome of *B. abortus*, and 110 for the second one (available at https://figshare.com/articles/Summary_of_whole_genome_sequencing_for_B_abortus/9747653). This high number of targets is consistent with GcrA being associated with the housekeeping sigma factor ($\sigma^{70}$), similarly to *C. crescentus*[60]. Among GcrA targets, several genes are involved in DNA repair (*ogt, lexA, uvrA,* and *mutL* but also *aidB*) (Fig. 6a). Compared to the rest of the chromosomes, we found a significantly higher (4.4- and 4.7-fold in chromosomes I and II, respectively) frequency of GANTC sites in the peaks of this ChIP-seq ($p < 0.001$ according to a Poisson distribution), which is consistent with methylation-dependent binding of GcrA in *B. abortus*.

To test if genes involved in DNA repair are regulated by GcrA, we constructed a strain with IPTG-inducible GcrA factor (Δ*gcrA* pBI-*gcrA*). In the absence of IPTG, the Δ*gcrA* pBI-*gcrA* strain mainly formed Y-shaped bacteria within the first 6 h of growth, indicating that their division was impaired. Texas Red succini-midyl ester (TRSE) labeling[40] also suggested that bacterial growth was also slower at later time points (Fig. 6b). After 3 h in the absence of IPTG, the bacteria were efficiently depleted from

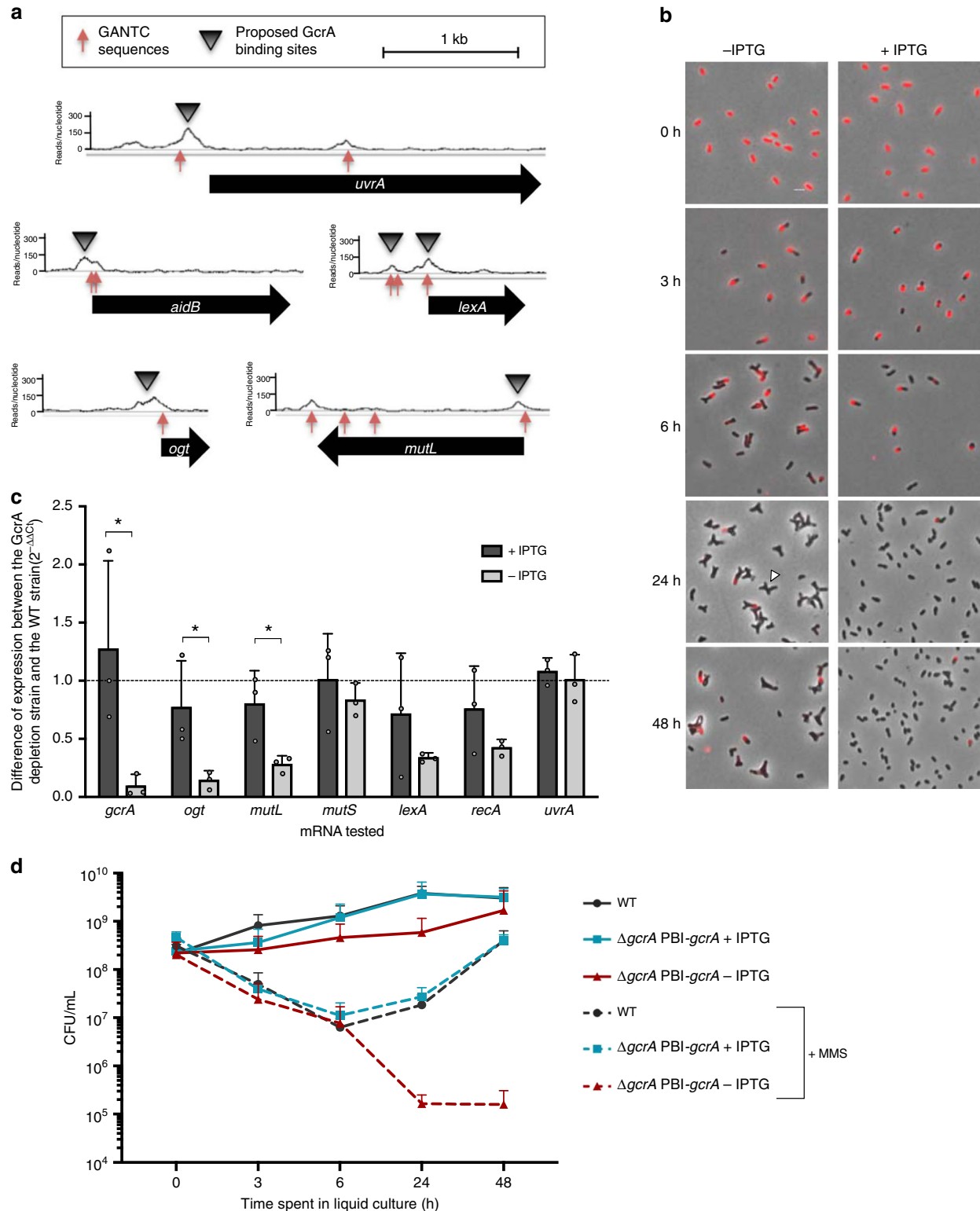

leftover GcrA, as demonstrated by western blot analysis (Supplementary Fig. 8a). RT-qPCR was performed on DNA repair genes after culturing *B. abortus* Δ*gcrA* pBI-*gcrA* in the presence or absence of IPTG. This confirmed that GcrA regulates the expression of *ogt* and *mutL* (Fig. 6c). Of note, the induction of *lexA* still occurred after MMS exposure in the absence of GcrA (Supplementary Fig. 8b), indicating that the activation of the SOS response under exogenous stress is regulated through a GcrA-independent mechanism.

The WT and GcrA-depleted strains were grown in liquid medium supplemented or not with IPTG and/or MMS to test whether the presence of GcrA is crucial for survival and growth in alkylating conditions. Aliquots were taken at different time points and plated on rich media supplemented with IPTG to assess bacterial survival. As seen with TRSE labeling (Fig. 6b), the Δ*gcrA* pBI-*gcrA* strain cultured without IPTG was almost not multi-plying but did survive (at least up to 2 days) as bacteria were recovered following plating on media supplemented with IPTG

**Fig. 6** Targets and functions of the transcription factor GcrA. **a** GcrA-binding sites detected by ChIP-seq. The number of reads per nucleotide is plotted for five promoter regions enriched by GcrA pull-down. **b** GcrA depletion generates growth and division defects in *B. abortus*. Bacteria were labeled with TRSE to covalently bind Texas Red to amine groups present at the bacterial surface. Non-labeled area thus correspond to newly incorporated envelope material. Grown in rich medium in the presence of IPTG (+IPTG), bacteria have a normal morphology. Upon IPTG removal (−IPTG), bacteria elongate (3 h), then form branches (6 h). At 24 h post IPTG removal, many bacteria present Y-shapes or more complex branched phenotypes (white arrow). **c** Gene expression in the GcrA-depleted strain. The mRNA levels of several genes coding for DNA repair proteins were calculated through RT-qPCR experiments for the GcrA-depleted strain. As predicted by ChIP-seq experiment, *ogt* and *mutL* expression are both affected by the absence of GcrA (−IPTG). The expression of the other genes was not statistically different (Student's *t*-test) between the two conditions (±IPTG) (*n* = 3). Source data are provided as a Source Data file. **d** Survival of GcrA-depleted strain in the presence of in vitro alkylating stress. Bacteria were cultured in liquid medium supplemented or not with IPTG and in the presence or absence of 5 mM of MMS. Samples were taken after 0, 3, 6, 24, and 48 h of culture and plated on rich medium supplemented with IPTG. Colony-forming units were counted to evaluate survival. Error bars represent standard deviation (*n* = 3). Source data are provided as a Source Data file

(Fig. 6d). When bacteria were cultured in the presence of MMS, both strains first underwent a strong drop of CFU, independent of IPTG. Subsequently, both the WT and the Δ*gcrA* pBI-*gcrA* strains were able to overcome the stress and recovered with time if supplemented with IPTG (Fig. 6d). When *B. abortus* was depleted of GcrA, bacteria were unable to recover (Fig. 6d), indicating that the presence of GcrA is required for *B. abortus* to efficiently cope with high exogenous alkylating stress.

**Individual repair pathways are required for long-term infections**. Three strains (Δ*ada*1 Δ*ada*2 Δ*ogt*, Δ*alkA* Δ*tagA*, and Δ*alkB*) which displayed increased sensitivity to alkylating stress in vitro (Fig. 4) were tested during the infection of RAW 264.7 macrophages but failed to show attenuation (Supplementary Fig. 9). Similarly, an *aidB* mutant had previously been shown to be unaffected in infection[63]. Other DNA repair mutants were also tested but none of them was attenuated in this model of infection (Supplementary Fig. 9).

According to RT-qPCR data, GcrA is involved in the regulation of at least two DNA repair pathways: direct repair through *ogt* and MMR through *mutL* (Fig. 6c). In RAW 264.7 macrophages, it was observed that the GcrA depleted strain maintained a stable number of CFU until 24 h post infection, before dropping at 48 h post infection (Fig. 7). This indicates that GcrA is required for survival in this model of infection. However, this attenuation might not be only due to the disruption of DNA repair pathways, as GcrA also regulates many other functions, which will require further investigation.

The Ada-based reporter system allowed us to determine that the alkylating stress occurring on *B. abortus* inside RAW 264.7 macrophages is very low. In addition, our assays have all focuses on early infection times. One possibility is that alkylation stress could be more important at later time points inside host cells and/or in more physiological infection conditions. For example, a *B. abortus* Δ*recA* mutant was previously shown to be attenuated in a mouse infection model[47] even though we could not detect any attenuation in RAW 264.7 macrophages (Supplementary Fig. 9). Therefore, we tested several mutant strains in an intranasal mice infection model[64] at 12 and 28 days (Fig. 8). In the lungs, there was a striking attenuation at both time points for the alkylation-specific direct repair Δ*ada*1 Δ*ada*2 Δ*ogt* mutant and the glycosylase (Δ*alkA* Δ*tagA*) deficient strain. The double endonuclease Δ*xthA*1 Δ*xthA*2 mutant and the MMR Δ*mutL* Δ*mutS* mutant were also strongly attenuated. In spleen, the defects were mild, with the BER mutants being the most affected (Fig. 8).

## Discussion

Many environmental and pathogenic bacteria possess an adaptive system against alkylating stress[1], indicating that this stress is widespread in the environment. Nevertheless, before this study, it was not known whether intracellular bacteria also face alkylation

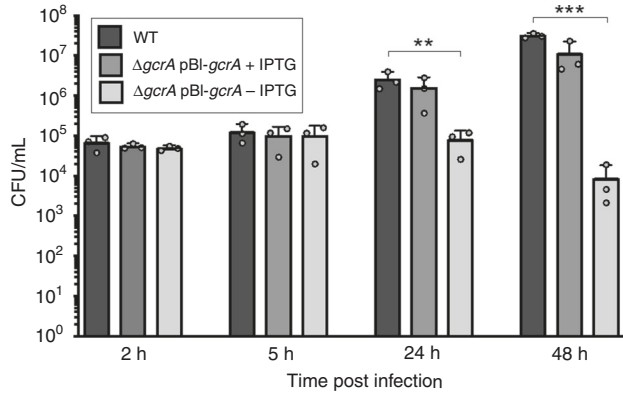

**Fig. 7** Infection of RAW 264.7 macrophages with the GcrA depleted strain. Colony-forming units were counted after 2, 5, 24, and 48 h post infection for the WT and the GcrA depleted strains incubated with or without IPTG (+IPTG or −IPTG, respectively). Error bars represent standard deviations (*n* = 3). A Scheffe statistical analysis reveals that, in the absence of IPTG, the GcrA-depleted strain is attenuated at 24 (*p* < 0.01, **) and 48 h (*p* < 0. 001, ***) post infection in this cell type. Source data are provided as a Source Data file

during infection. Here, we show that the intracellular pathogen *B. abortus* faces alkylation stress during infection, with functional DNA repair pathways for alkylation damage required in a mice model of infection, and that the control of the genes involved in the repair of alkylated DNA is different from the one reported in *E. coli*.

To investigate the occurrence of alkylating stress at early time points of infection, we developed a fluorescence-based reporter system to follow the occurrence of the stress on bacterial DNA at the single-cell level. To understand the source of alkylating stress, a probe was designed to be covalently attached at the bacterial surface and report N-nitrosation events occurring inside the eBCV. The combination of those two approaches allowed us to determine that alkylating stress occurs inside the eBCV mainly via the bacterial metabolism. Indeed, the addition of ascorbate, which quenches N-nitrosation on the bacterial surface, was not able to decrease alkylation damage detected with the Ada-based reporter. Moreover, the deletion of *moaA*, which is involved in the biosynthesis pathway of the molybdenum cofactor, decreased the intensity of alkylating stress. *B. abortus* possesses a single nitrate reductase, so it is likely that the phenotype of the *moaA* mutant comes from the simultaneous deficiency of Nar and other enzymes dependent on the molybdenum cofactor[7]. It could be the case of MSF, a nitrate–nitrite antiporter, as well as FdnG, a formate dehydrogenase involved in nitrate respiratory chain[65]. Importantly, the occurrence of external N-nitrosation events is relevant for alkylating stress only if metabolites are present in the

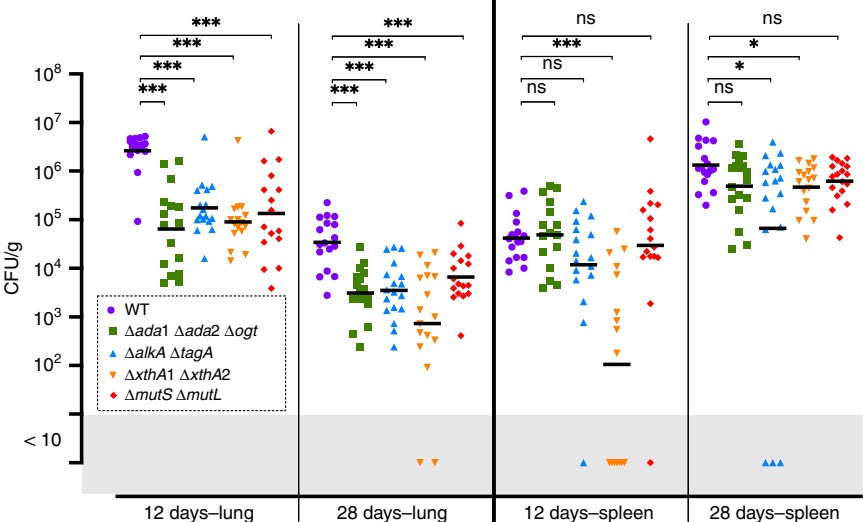

**Fig. 8** Intranasal mice infection with *B. abortus* DNA repair deletion strains. Wild-type C57BL/6 mice received $2 \times 10^4$ CFU of *B. abortus*, as indicated in the section "Methods". The mice were sacrificed at the selected time post infection. The data represent the number of CFU $g^{-1}$ of lung and spleen. These results are representative of two independent experiments with, in the same order than in the figure, $n = 9 + 8, 8 + 8, 9 + 8, 8 + 8, 8 + 8$ mice at 12 days and $n = 9 + 8, 9 + 9, 9 + 9, 9 + 8, 9 + 9$ mice at 28 days. Black lines correspond to mean values. A Mann–Whitney test was performed with $p > 0.05$ (ns non significant), $p < 0.05$ (*) and $p < 0.001$ (***). Source data are provided as a Source Data file

environment. Indeed, alkylating stress is generated by those modified metabolites, and not by N-nitrosation per se[66]. It has been proposed that the eBCV is deficient in metabolites[67,68], which could explain why host cells do not produce detectable exogenous alkylating agents. Another important point is that N-nitrosation reactions are dependent on RNS levels[25] and *B. abortus* is known to be very weakly immunogenic, as it prevents proinflammatory responses in macrophages and neutrophils[69]. In addition, *B. abortus* possesses a gene coding for a nitric oxide reductase, involved in detoxification of NO and nitrate respiration, which is required and overexpressed by the bacterium early during infection[68,70,71]. One interesting hypothesis is that by avoiding exogenous sources of alkylating stress, *B. abortus* generates weak endogenous stress. Indeed, as *B. abortus* activates its nitrate respiration/denitrification pathway inside host cells, it must probably also generate endogenous N-nitroso compounds and hence endogenous alkylating agents. Importantly, alkylating stress does not induce a linear dose response in organisms[72], which could explain why DNA repair mutants were not attenuated during macrophage infection, but was attenuated in the more physiologicaly and relevant intranasal mouse infection model.

*E. coli* deals with alkylating stress through various and dynamically regulated DNA repair pathways. Indeed, in the model developed by Uphoff[34], *E. coli* mainly relies on damage tolerance (via the SOS response) and constitutive repair (via TagA and Ogt), before the adaptive response takes over after prolonged alkylating stress. Our data indicate that *B. abortus* has a different strategy to cope with this stress. First, we found that an Ada-based adaptive system is absent in *B. abortus*, similarly to some other α-proteobacteria[59,73]. Nevertheless, *B. abortus* does possess the ability to induce the expression of both *alkA* and *tagA* upon alkylating agent exposure, via a yet unknown mechanism. Secondly, we showed that, through the SOS response, the genes coding for the error-prone DNA polymerase *imuABC* were also overexpressed in conditions of high alkylating stress. Finally, *B. abortus* was found to rely on the essential and well-conserved transcription factor GcrA to control the expression of a series of genes involved in DNA repair, including *mutL* and *ogt*. The GcrA depleted strain was impaired for division, growth, and virulence,

suggesting that it plays a role in *B. abortus* cell cycle regulation. Of note, in *B. abortus*, the promoter of *tagA* is directly bound by CtrA, another conserved cell cycle regulator[57]. In *E. coli*, *ada* is known to be overexpressed in the stationary phase, independent of the methylation of its C38 residue but through the activity of the alternative sigma factor RpoS[7]. Since there appears to be no *rpoS* homolog in *B. abortus* and other α-proteobacteria[74], it is tempting to speculate that these bacteria have selected systems in which cell cycle regulators control DNA repair. A previous report[53] also indicates that genes of the SOS regulon have a high basal expression in *B. abortus*, which could be an additional way to ensure sufficient protection against endogenously produced stress. Interestingly, the absence of genome replication inside the eBCV also constitutes a direct advantage against genotoxic stresses, as DNA adducts do not fix mutations so long as replication has not occurred. Another characteristic of GcrA in *C. crescentus* is its ability to sense CcrM-dependent methylation on DNA[61]. Knowing that this epigenetic mark is cell cycle regulated in *C. crescentus*[75] and probably also in *B. abortus*[57,58], there could be a functional link between both damage-induced and epigenetic methylation.

This study provides new insights into a stress that was, until now, only hypothetically associated with intracellular bacterial pathogens. The discovery of alkylating stress on intracellular *B. abortus* suggests that other bacteria transiting through similar compartments could also be exposed to such a stress. Future investigations along these lines could generate a better understanding of host–pathogen interactions at the molecular level.

## Methods

**Bacterial strains and media**. *E. coli* strains DH10B (Thermo Fisher Scientific) and S17-1 (ref. [76]) were grown in Luria-Bertani (LB) medium at 37 °C. *B. abortus* 544 Nal[R] strain (Obtained from J.-M. Verger and M. Grayon, Institut National de la Recherche Agronomique, Laboratoire de Pathologie Infectieuse et d'Immunologie, Nouzilly, France) and its derivatives were grown in either 2YT-rich medium (1% yeast extract, 1.6% peptone, 0.5% NaCl) or TSB-rich medium (3% Bacto tryptic soy broth) at 37 °C. Antibiotics were used at the following concentrations: ampicillin, 100 µg mL$^{-1}$; kanamycin, 10 µg mL$^{-1}$ with integrative plasmids or 50 µg mL$^{-1}$ with replicative plasmids; chloramphenicol, 20 µg mL$^{-1}$; nalidixic acid, 25 µg mL$^{-1}$; rifampicin, 20 µg mL$^{-1}$; gentamicin, 10 or 50 µg mL$^{-1}$ as indicated. When required, isopropyl β-D-1-thiogalactopyranoside (IPTG) was used at a concentration of 1 mM in bacterial culture and at 10 mM in the culture medium during cellular infections.

*B. abortus* deletion strains were constructed by allelic exchange, via pNPTS138 vectors (M. R. K. Alley, Imperial College of Science, London, UK) carrying a kanamycin resistance cassette and a sucrose sensitivity cassette[27]. Briefly, we selected bacteria which had integrated the plasmid containing upstream and downstream sequences (about 750 bp each) of our target gene by plating them on kanamycin-containing agar plates, then performed a counterselection on 5% sucrose-containing agar plates without kanamycin to allow plasmid curing. To confirm gene deletion, PCR was performed on colonies that were kanamycin-negative and sucrose-resistant with primers targeting sequences upstream and downstream of the plasmid-containing sequences. Note that in the case of the ΔxthA2 strain, 204 nucleic acids were kept on each side of the gene, as there was a previous report that the full deletion of the gene was not feasible[45]. Primers, plasmids and ORF of the studied genes are listed in Supplementary Data 1.

*B. abortus* GcrA-depleted strain was constructed similarly than the CtrA-depleted strain[57]. Briefly, the $p_{lacI}$-$lacI$-$p_{lac}$ sequence was amplified from the pSRK-Kan plasmid[77] using Phusion High-Fidelity DNA Polymerase (New England BioLabs). The PCR product was then cloned into a pBBRMCS1 plasmid using *Sac*I and *Bam*HI restriction enzymes. This modified pBBRMCS1 is referred to as pBI. The *gcrA* coding sequence was amplified form *B. abortus* 544 genome with Phusion High-Fidelity DNA Polymerase (New England BioLabs) and then cloned into pBI using *Bam*HI and *Kpn*I enzymes in order to orient the insert opposite to the $p_{lac}$ promoter already present in pBBRMCS1. This final plasmid (pBI-*gcrA*) was transferred to *B. abortus* by mating, after inserting the deletion plasmid (pNPTS138-ΔgcrA), then *gcrA* was removed from the chromosome of *B. abortus* as described above.

The *lexA* sequence was amplified from *B. abortus* 544 genome with primers listed in Supplementary Data 1 and the PCR product was ligated into a pBBRMCS1 plasmid after *Apa*I and *Bam*HI restriction, generating the pBBRMCS1-$p_{lac}$-*lexA* final plasmid.

**Cloning of the reporter system for alkylating stress**. The $p_{ada}$-$ada_{E.\ coli}$ sequence, including the start codon of $alkB_{E.\ coli}$, was amplified from *E. coli* DH10B with Phusion High-Fidelity DNA Polymerase (New England BioLabs) using primers listed in Supplementary Data 1. A superfolder *gfp* coding sequence (Supplementary Note 1), with a *Xho*I sequence after the start codon and a *Pst*I sequence after the stop codon, was adapted to fit the codon usage of *B. abortus* 2308 (http://www.kazusa.or.jp/codon/) and ordered as gBlocks gene fragment (Integrated DNA Technologies). Both DNA products were cloned into a pBBRMCS1 plasmid using *Pst*I, *Xho*I, and *Spe*I restriction enzymes in a triple ligation to orient the $p_{ada}$-$ada_{E.\ coli}$-$gfp$ fusion opposite to the $p_{lac}$ promoter of pBBRMCS1.

**Synthesis of N-nitrosation-sensitive probe and binding conditions**. The probe was designed based on Mio et al.[41], with the addition of a succinimidyl ester group in order to allow the binding of the probe on amines at the bacterial surface. The characterization of the probe can be found in Supplementary Note 2.

One milliliter of bacteria (DO$_{600}$ 0.5) was centrifuged at 7000 r.p.m. for 2 min and washed twice in phosphate buffered saline (PBS). They were incubated for 1 h at 37 °C with the probe (10 μM) in 1 mL of PBS supplemented with 100 μL of NaHCO$_3$ 1 M (pH 8.4). Bacteria were then washed three times with PBS and used either for RAW 264.7 infection or for experiments in culture. For experiments in culture, labeled bacteria were left for 1 h on wheel in the dark with 1 M of KNO$_2$ (Thermo Fisher scientific) and 20 μL of HCl 3 M, before to be washed twice with PBS and fixed with paraformaldehyde (PFA) 2% for 20 min at 37 °C.

**Texas Red succinimidyl ester labeling**. One milliliter of bacteria (DO$_{600}$ 0.5) was centrifuged at 7000 r.p.m. for 2 min and washed twice in PBS. Bacteria were resuspended in 1 mL of PBS and incubated with Texas Red succinimidyl ester (TRSE) at a final concentration of 1 μg mL$^{-1}$ (Invitrogen) for 15 min at room temperature. Bacteria were then washed three times with PBS.

**RAW 264.7 macrophage culture and infection**. RAW 264.7 macrophages (ATCC) were cultured at 37 °C in the presence of 5% CO$_2$ in DMEM (Invitrogen) supplemented with 4.5 g L$^{-1}$ glucose, 1.5 g L$^{-1}$ NaHCO$_3$, 4 mM glutamine, and 10% fetal bovine serum (Gibco). RAW 264.7 macrophages were seeded in 24-well plates (with coverslips for immunolabeling) at a concentration of 10$^5$ cells per mL and left in the incubator overnight. The next morning, late exponential phase cultures of *Brucella* (DO$_{600}$ 06–0.9) were washed twice in PBS in order to remove antibiotics and traces of growth medium, then they were prepared in DMEM at a multiplicity of infection of 50. During that step, IPTG or ascorbate was added to the culture medium if required. Bacteria and cells were centrifuged at 400 *g* for 10 min at 4 °C and then incubated for 1 h at 37 °C with 5% CO$_2$ atmosphere before to be washed twice with PBS and then incubated in medium supplemented with 50 μg mL$^{-1}$ of gentamicin to kill extracellular bacteria. One hour later, the medium was replaced by fresh medium supplemented with 10 μg mL$^{-1}$ of gentamicin.

**Immunolabeling of infected RAW 264.7 macrophages**. Cells were washed twice in PBS before to be fixed for 20 min in 2% PFA pH 7.4 at 37 °C. They were then left in PBS in the dark at 4 °C overnight before to be permeabilized in PBS with 0.1% Triton X-100 (Prolabo) for 10 min. Cells were incubated for 45 min with primary

antibodies in PBS containing 0.1% Triton X-100 and 3% (w/v) bovine serum albumin (BSA, Sigma-Aldrich). Next, cells were washed three times in PBS before to be incubated with secondary antibodies in PBS containing 0.1% Triton X-100 and 3% BSA. For LampI labeling experiments, the primary antibodies consisted in homemade anti-*Brucella* rabbit polyclonal antibodies[27] and anti-LampI rat antibodies (1D4B; Developmental Studies Hybridoma Bank, University of Iowa), and the secondary antibodies consisted in goat anti-rabbit antibodies coupled to Pacific Blue (Invitrogen, cat. no. P10994) and goat anti-rat antibodies coupled to Alexa Fluor 647 (Invitrogen, cat. no. A21247). For all other experiments, we used anti-*Brucella* LPS primary antibodies (A76-12G12, undiluted hybridoma culture supernatant[78]) and goat anti-mouse secondary antibodies coupled to Texas Red (1:500) (Invitrogen, cat. no. T862). Coverslips were washed three times in PBS, once in ddH$_2$O and then mounted on Mowiol (Sigma).

**Microscopy and analyses of fluorescence**. We used a Nikon Eclipse E1000 (objective ×100, plan Apo) microscope connected to a ORCA-ER camera (Hamamatsu). The Hg lamp was set with ND filter at 4. Bacteria in culture (2 μL) were observed with the phase contrast on PBS-agarose (1%) pads. Bacteria inside host cells were observed with the TxRed channel (or the CFP channel for LampI experiments) (100 ms). The FITC channel (1 s) was used to detect either the N-nitrosation-sensitive probe or the GFP signal of the reporter system. LampI proteins were detected with the APC channel (800 msec). Pictures were encoded with NIS-element software and analyzed with the plug-in MicrobeJ in ImageJ[79]. For bacteria on pads, mean fluorescence intensities (MFI) were obtained as the "mean_c" values with MicrobeJ for individual bacteria. For intracellular bacteria, MFI were obtained by subtracting the background fluorescence (defined here as the average value of fluorescence given by the Pixel Inspection Tool on three points randomly chosen around a bacterium) to the "mean" value of fluorescence obtained for each bacterium with MicrobeJ (see Supplementary Note 3 for a detailed protocol). Note that MFI values were considered as positive for bacteria labeled with the N-nitrosation-sensitive probe when they reached an arbitrary threshold of MFI = 100, as this value is above 86% of the MFI values on labeled bacteria (in blue in Supplementary Fig. 5a), and below 96% of the MFI values for the positive control (labeled bacteria + KNO$_2$, in orange in Supplementary Fig. 5a).

**Chromatin immunoprecipitation with anti-GcrA antibodies**. Cultures of 80 mL of *B. abortus* (OD$_{600}$ 0.8) were harvested by centrifugation and proteins were cross-linked to DNA with 10 mM sodium phosphate buffer (pH 7.6) and 1% (v/v) formaldehyde for 10 min at RT and 30 min on ice. Bacteria were centrifuged and washed twice in cold PBS before to be resuspended in lysis buffer (10 mM Tris-HCl pH 7.5, 1 mM EDTA, 100 mM NaCl, 2.2 mg mL$^{-1}$ lysozyme, 20 mL protease inhibitor solution from Roche). Bacteria were lysed, after the addition of 0.1 and 0.5 mm diameter Zirconia/Silica beads (Biospec Products), in the cell Disruptor Genie from Scientific Industries at maximal amplitude (2800) for 25 min at 4 °C. Bacteria were then incubated for 10 min in the presence of ChIP buffer (1.1% Triton X-100, 1.2 mM EDTA, 16.7 mM Tris-HCl pH 8.0, 167 mM NaCl, protease inhibitors). DNA fragments of about 300 base pairs were obtained by sonicating the lysate on ice (Branson Sonifier Digital cell disruptor S-450D 400 W) by applying 15 bursts of 20 s (50% duty) at 30% amplitude. Debris were excluded in the pellet by centrifugation at 14,000 r.p.m. for 3 min. The supernatant was normalized by protein content by measuring the absorbance at 280 nm and 7.5 mg of protein was diluted in 1 mL of ChIP buffer supplemented with 0.01% SDS and pre-cleared in 80 μL of protein A-agarose beads (Roche) and 100 μg BSA. Homemade anti-rabbit polyclonal GcrA antibodies (290.S3) were added to the supernatant (1:1000) and incubated overnight at 4 °C. The mix was then incubated with 80 mL of protein A-agarose beads pre-saturated with BSA for 2 h at 4 °C. Beads were then washed in the following order: once with low salt buffer (0.1% SDS, 1% Triton X-100, 2 mM EDTA, 20 mM Tris-HCl pH 8.1, 150 mM NaCl), once with high salt buffer (0.1% SDS, 1% Triton X-100, 2 mM EDTA, 20 mM Tris-HCl pH 8.1, 500 mM NaCl), once with LiCl buffer (0.25 M LiCl, 1% NP-40, 1% sodium deoxycholate, 1 mM EDTA, 10 mM Tris-HCl pH 8.1), and twice with TE buffer (10 mM Tris-HCl pH 8.1 and 1 mM EDTA) before to be eluted with 500 μL of elution buffer (1% SDS and 0.1 M NaHCO$_3$). The reverse-crosslinking was performed with 500 mL of 300 mM of NaCl overnight at 65 °C. Samples were then treated with Proteinase K (in 40 mM EDTA and 40 mM Tris-HCl pH 6.5) for 2 h at 45 °C and DNA was finally extracted with QIAGEN MinElute kit to be resuspended in 30 μL of Elution buffer.

Illumina MiSeq was used to sequence immunoprecipitated DNA. Data consisted of a number of reads per nucleotide. A Z-score for each base pair (i.e. the number of standard deviations from the average) was calculated based on average and variance in a window of 1 million base pairs. A threshold of Z-score above 4 was set to consider genomic regions as bound by GcrA. These sequences were mapped to the genome of *B. abortus* 2308 (available at https://www.ncbi.nlm.nih.gov/geo/query/acc.cgi?acc = GSE136733). The GcrA-binding peaks (.txt files) can also be visualized on Artemis (freely available at http://www.sanger.ac.uk/science/tools/artemis) with the genomic sequences (.gb files) available at https://figshare.com/s/0e580305b65f67619d36. To calculate the number of GANTC sequences in ChIP-seq peaks, we extracted peak sequences online with Emboss-extractseq (http://emboss.bioinformatics.nl/cgi-bin/emboss/extractseq) and looked for the presence of GANTC sites with the "pattern matching, dna-pattern" tool on

RSATools[80] (http://embnet.ccg.unam.mx/rsa-tools/) on both strands and with allowing overlapping matches. Results were normalized according to peak size to obtain the number of GANTC sites per kb (GANTC/kb). A similar analysis was performed for whole chromosomes with the "pattern matching, genome-scale dna-pattern" tool on RSATools. Ratios were calculated between data obtained (in GANTC/kb) for the peaks and for the whole chromosomes.

**Mouse infection**. *Ethics statement*: The Animal Welfare Committee of the Université de Namur (UNamur, Belgium) reviewed and approved the complete protocols for *Brucella* infections (Permit Numbers: 05-558 for intraperitonneal infections and 19-330 for intranasal infections).

Mice were acquired from Harlan (Bicester, UK) and bred in the animal facility of the Gosselies campus of the Université Libre de Bruxelles (ULB, Belgium). For intraperitoneal infections, mice (C57BL/6, 10–12 weeks old females) were injected with a dose of $10^5$ CFU per mL of *B. abortus* in 500 μL of PBS. Infectious doses were validated by plating serial dilutions of inoculums. At 60 h post inoculation, mice were euthanized by cervical dislocation. Immediately after being killed, spleens were recovered in PBS with 0.1% Triton X-100 (Sigma) and plated on 2YT agar medium.

For intranasal infections, mice (C57BL/6, 10–12 weeks old, mix of males and females) were anesthetized with a mix of Xylasine (9 mg kg$^{-1}$) and Ketamine (36 mg kg$^{-1}$) in PBS before being inoculated by an intranasal injection of $2 \times 10^4$ CFU mL$^{-1}$ of *B. abortus*. Confirmation of the infectious doses was done by plating serial dilutions of the inoculums. Mice were sacrificed by cervical dislocation at 12 or 28 days post infection. Immediately after sacrifice, spleen and lungs were crushed and resuspended in PBS 0.1% Triton X-100 (Sigma-Aldrich). The bacterial load was evaluated by serial dilutions in RPMI and plated on TSB agar medium. The CFU were counted after 5 days of incubation at 37 °C. All experiments were done in a Biosafety level 3 facility.

**Whole-genome sequencing after infection and liquid cultures**. Two liquid cultures of WT *B. abortus* were inoculated in 2YT-rich medium from the same plate. One of them was divided into five subcultures and diluted (1:10) in liquid cultures twice every 24 h, before to be plated. The remaining original liquid culture was used to infect five C57BL/6 mice and RAW 264.7 macrophages. Mice were injected intraperitoneally as described above and were euthanized at 60 h post inoculation to recover spleens. RAW 264.7 macrophages were infected as described above and bacteria were plated at 6 and 48 h post infection. Five streaks were made from five isolated colonies obtained after passage in either liquid cultures, mice, or RAW 264.7 macrophages from different wells. The five streaks served for inoculation of liquid cultures, from which genomic DNA was extracted (NucleoSpin Tissue extraction kit, Macherey-Nagel). Samples were sequenced with the Illumina sequencing technique using NextSeq500 run Mid PE150 after preparing a TruSeq DNA library (performed by Genomics Core Leuven, Belgium). Sequencing hits were mapped on the genome of *B. abortus* 544 and, for each colony, all the nucleotides that were different from the reference strain were compiled in an Excel table (performed by Genomics Core Leuven, Belgium; data available at https://figshare.com/articles/Summary_of_whole_genome_sequencing_for_B_abortus/9747653). Mutations were counted for each isolated colony by excluding regions corresponding to microsatellites and with less than 10 reads.

**Reverse transcription followed by quantitative PCR**. Bacterial cultures were grown in rich medium to exponential phase (OD$_{600}$ 0.3), washed twice in PBS, and allowed to grow in rich medium supplemented or not with IPTG and/or MMS 2.5 mM (Sigma) for 5 h. Bacteria were washed twice in PBS, then collected by centrifugation, and immediately frozen and stored at −80 °C until processing. RNA was extracted using TriPure isolation reagent (Roche) according to the manufacturer's instructions. Samples were treated with DNase I (Fermentas), then RNA was reverse transcribed with specific primers (Supplementary Data 1), using the High capacity cDNA Reverse Transcription kit (Applied Biosystems). Specific cDNAs were amplified on a LightCycler 96 Instrument (Roche) using FastStart Universal SYBR Green Master (Roche). Results were normalized using 16S RNA as a reference with the $E^{-\Delta\Delta Ct}$ calculation method[81], where $E$ corresponds to the efficiency of the primers (determined by serial dilution).

**Colony-forming units counts**. For CFU counts after infection, RAW 264.7 macrophages were washed twice in PBS, then lysed with 0.01% Triton X-100 in PBS for 10 min at 37 °C. Several dilutions were plated on TSB supplemented with IPTG when required. Plates were incubated for 3 days at 37 °C.

For CFU counts in culture, wild-type *B. abortus* and GcrA-depleted strain supplemented with IPTG were grown to the mid exponential phase (OD$_{600}$ 0.3–0.6) and normalized to OD$_{600}$ 0.15. Cultures were divided into different aliquots to be tested with or without MMS 5 mM and with or without IPTG. Samples were taken at different time points (3, 6, 20, 24, and 48 h) and plated with serial dilutions on 2YT plates, supplemented with 1 mM IPTG for the depleted strain.

For CFU counts on plates supplemented with MMS or MNNG, 100 μL of bacteria in the mid exponential phase (OD$_{600}$ 0.3–0.6) were plated after normalization to OD$_{600}$ 0.1 of all bacterial cultures. All plates were prepared fresh, 1–2 h before each experiment. MMS was added at a concentration of 2.5 mM in TSB-agar-rich medium during plate preparation. MNNG was prepared in an

acetate buffer (pH 5), and then added at a final concentration of 35 μM in TSB-agar-rich medium during plate preparation.

**Western blot**. Cultures of *B. abortus* in the late exponential phase (OD$_{600}$ 0.7–1) were concentrated to an OD$_{600}$ of 10 in PBS, then inactivated for 1 h at 80 °C. Loading buffer (1:4 of final volume) was added before to heat the sample at 95 °C for 10 min. Ten microliters of sample were loaded on each well of a 12% acrylamide gels. After migration, proteins were transferred with the semi-dry method onto a nitrocellulose membrane (GE Healthcare) which was blocked in PBS supplemented with 0.05% Tween 20 (VWR) and 5% (w/v) milk (Nestlé, foam topping) overnight. The membrane was incubated for 1 h with polyclonal anti-GcrA (290.S3) or monoclonal anti-Omp10 (A68/4B10/F05) primary antibodies (1:1000), then with secondary antibodies coupled to HRP (Dako Denmark) (1:5000), both in PBS 0.05% Tween 1% milk. The membrane was washed three times in PBS before to be revealed with The Clarity Western ECL Substrate (Biorad) and Image Quant LAS 4000 (General Electric).

**Statistical analysis**. Statistical tests were performed in Excel or in GraphPad Prism version 8.0. All statistical tests were one-sided.

**Reporting summary**. Further information on research design is available in the Nature Research Reporting Summary linked to this article.

## Data availability
ChIP-seq data are available at the GEO database under accession code GSM4056904. The list of mutations for whole genome sequencing experiments is available at figshare [https://figshare.com/articles/Summary_of_whole_genome_sequencing_for_B_abortus/9747653]. The source data underlying Figs. 2b, 3b-c, 4, 5, 6c-d, 7, and 8 and Supplementary Fig. 3b-d, 5a-b, d, 8a-b, 7, and 9 are provided as a Source Data file. All other relevant data supporting the key findings of this study are available within the article and its Supplementary Information file or from the corresponding author upon request.

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

## Acknowledgements

We thank members of the URBM and I. Matic for stimulating discussions. We also thank Y. Ashhab for his help regarding ChIP-seq analysis. We thank H. Lavender and C. M. Tang for their careful reading of our manuscript. We thank UNamur (https://www.unamur.be/) for financial and logistic support. This work was funded by The Fédération Wallonie-Bruxelles (ARC 17/22-087) and by the FRS-FNRS "Brucell-cycle" (PDR T.0060.15). K.P., A.M., and G.P. were supported by a FRIA Ph.D. fellowship. A.R. and N.F. held an Aspirant fellowship from FRS-FNRS. This work was also supported by the French Agence Nationale de Recherche (ANR-JCJC-2011-Castacc) (http://www.agence-nationale-recherche.fr/) and the Region Pas-De-Calais (http://www.nordpasdecalais.fr) CPER to A.F. and E.G.B.

## Author Contributions

K.P. and X.D.B. wrote the manuscript. R.J. and S.P.V. created and characterized the N-nitrosation sensitive probe. A.M., G.P. and E.M. designed and performed mice infection experiments. K.P. and X.D.B. designed all other experiments, and E.G.B. contributed to the design of the ChIP-seq experiment. K.P., A.R., G.P., A.F., N.F., K.W. and N.Z. performed experiments.

## Competing interests

The authors declare no competing interests.
