## [Peer Review File · Nature Communications]

Reviewers' comments:

Reviewer #1 (Remarks to the Author):

With the exception of a few specific sections, which I will point out below, this is a well-written paper describing a very thorough and creative study that provides convincing evidence that *Brucella* strains experience alkylating stress during their intracellular residence in mammalian cells. The authors also demonstrate that these bacteria employ a mechanism unlike that of *E. coli* (which is often the paradigm of bacterial DNA repair studies) for dealing with the DNA damage caused by alkylation stress and identify the specific gene product involved in this repair. Additionally, they demonstrate that the global transcriptional regulator GcrA plays an important role in modulating the response of *Brucella* to alkylating stress. Overall, these studies provide an important contribution to our understanding of the basic biology of an important zoonotic pathogen and will have a significant impact on the *Brucella*, alpha-proteobacteria and prokaryotic DNA repair fields. But I am not convinced that they tell us a lot about the contribution of specific DNA repair pathways to *Brucella* virulence.

The only specific concern about the present version of the manuscript is that in at least two places – e.g. lines 22-27 in the Abstract and lines 376-390 in the Discussion section, the authors should re-read their statements and revise the grammar so that the reader will understand the points that they are trying to make. Although the manuscript is in general well-written, this reviewer was confused by both of these sections.

The only general concern I have about the manuscript has to do with the 'virulence' studies employing the RAW264.7 cells. Although this cell line has been widely used to study the intracellular life cycle of *Brucella* strains, the replication properties in these bacteria in RAW cells do not always mimic their intracellular replication in primary explant macrophages. In fact, when one considers the link between NO, RNS and alkylation stress, it would seem more fitting to examine the intracellular replication profiles of *B. abortus* DNA repair mutants in primary explant macrophages stimulated in interferon gamma (which stimulates their iNOS activity). This reviewer would also contend that examining the spleen colonization profiles of the *B. abortus* mutants in mice would be an even more relevant way of determining the contributions of the individual gene products to virulence. It was previously shown that a *B. abortus* *recA* mutant displays significant attenuation in the BALB/c mouse model of chronic infection (Tatum et al. 1993. *Microb. Path.* 14: 177-185), and this point needs to be mentioned in this paper and taken in consideration when the authors provide their interpretations for the data presented in Figure S8.

Reviewer #2 (Remarks to the Author):

Poncin et al. studied whether *B. abortus* experiences alkylation stress during infection and how alkylated DNA damage is repaired in this bacterium. They developed an N-nitrosation sensitive fluorescent probe and tested the survival of several mutants of *B. abortus* lacking genes involved in DNA alkylation repair. The findings are potentially interesting and novel & it is important to understand molecular mechanisms in understudied pathogens such as *B. abortus*. However, there are several issues with the work as presented in this manuscript. Additional experiments and controls will help to solidify the findings and increase the significance of this work. Specific points:

1. The N-nitrosation-sensitive probe for use in live cells is potentially a useful tool. However, crucial data and controls are missing to demonstrate the sensitivity and robustness of the method. The only data provided are in Fig. S5 and Fig. 3 showing bar plots of median fluorescence or percentage of cells with elevated fluorescence. The raw fluorescence microscopy snapshots and fluorescence distributions should be shown as well, not just the bulk values. How was the percentage of positive cells determined? Was a threshold intensity used, and if so, how was the

threshold determined? Are the 23% of cells a genuine subpopulation in a bimodal distribution of fluorescence, or are these cells just the tail of a single distribution? More information has to be provided in the Materials and Methods how imaging was done for this probe. Further to this point, in the discussion section it is written (Line 356): "This technique is particularly interesting because it allows the examination of the environment of each bacterial cell at a given time. Ultimately, this single-cell fluorescent approach could be further extended to other questions,..." However, no analysis on the level of single bacteria or examination of the environment was shown in this work.

2. One of the key conclusions of this paper is that *B. abortus* experiences "alkylation stress" during infection. However, it is not shown that the cells actually suffer "stress". The Ada fluorescence assay and the N-nitrosation probe indicate the presence of alkylating agents but do not show that the cells are "stressed" by this. In fact, there is no change in mutagenesis during infection (Fig. S4) – alkylating agents should have left a specific mutation signature. There is also no change in CFU during infection with *B. abortus* strains that are defective in DNA alkylation repair (Fig. S8).
3. The authors argue that alkylation occurs specifically in the eBCV. However, this was inferred only indirectly from the time post infection. The authors need to demonstrate that the majority of *B. abortus* indeed reside in the eBCV at 5 hours PI. Especially since only 23% of cells show elevated N-nitrosation, it is important to know what fraction of cells is inside the eBCV.
4. The authors show mean fluorescence ratios of the Ada reporter in Fig. 2. However, given previous reports of Ada heterogeneous expression, it would be important to show the distribution of expression at a single cell level. Is expression limited to a small fraction of cells like the 23% of N-nitroso positive cells?
5. The link between GcrA and response to alkylation stress remains quite opaque. (i) As it is written, it is unclear if methylation by alkylation stress can affect GcrA activity, and thus the expression of target genes. (ii) Depletion of *gcrA* clearly has a pleiotropic effect on cell fitness. Therefore, infection of macrophages with a *gcrA* depletion strain does not really inform whether the decrease of cell survival during infection is actually due to a deficiency in DNA repair mechanisms. (iii) Fig. 5 shows that only AlkA and Tag are induced after MMS treatment (apart from the known SOS genes), but the mechanism of induction of these genes was not further investigated or discussed. (iv) Following from point iii, why did the authors not use *B. abortus* endogenous AlkA as a reporter for induction of the alkylation response, instead of using the artificial *E. coli* Ada reporter?
6. I find the discussion section lacks focus and the writing could be much improved (I included a few suggestions in the minor points below). The authors present numerous suggestions for further experiments and speculations that are not directly supported by any experiments in this manuscript. Instead, they should concisely summarize the key findings and explain their significance.

Minor points:

1. It is surprising that *aidB* was not included in the knock out (Fig. 4) and expression (Fig. 5) experiments, since it is conserved and clearly linked to alkylation stress.
2. The alkylating agent concentrations used in vitro were much higher than those detected in vivo. For reference, can the authors show at what MMS concentration in vitro they detect similar Ada activation as in vivo, and subsequently compare cell survival at that concentration?
3. Line 195: "The addition of ascorbate strongly decreased the mean fluorescence intensity (MFI) of labeled bacteria (Fig 3B)." This sentence probably refers to Fig. 3C not 3B.
4. Line 326: Usually MMR is used as abbreviation for mismatch repair.
5. Line 362: "For example, it could occur through waves with some bacteria being affected only at a given time point." This is very speculative. What is the evidence for "waves"?
6. Line 387: "The circle is now complete:..." This is a strange expression, and should be revised
7. Line 423: "Indeed, DNA adducts do no cause any mutation as long as replication has not occurred." The statement should be revised as it is not true as written. For example, stress induced mutagenesis occurs in stationary phase cultures.
8. Line 458: "...so it is a good indication that this bacterium probably meets the stress in the intracellular environment." The simultaneous use of "good indication" and "probably" is

contradicting.

9. Line 460: "Future research on this matter could be of great interest for the development of novel antimicrobial therapies." This statement is of little value unless additional information is provided as to what such novel therapies might involve, and how the findings in this manuscript relate to it.

10. Figure 2B and 3C show the median of biological triplicates. For such small number of repeats, the mean would be a more descriptive and reliable statistic. Usually, the median is used for skewed distributions, whereas experimental repeats should follow a symmetrical distribution, i.e. best summarized by the mean.

Reviewer #3 (Remarks to the Author):

The manuscript entitled "Occurrence and repair of alkylating stress in the intracellular pathogen *Brucella abortus*" by Prof De Bolle and colleagues describes the phenomenon of the response of intracellular bacterium *Brucella abortus* to alkylating agents occurring endogenously (produced by *B. abortus*) and exogenously (inside the host cell). The work is interesting but I am not sure if Authors fully manage the subject. I have a feeling that they do not take under consideration alkylation (methylation) processes essential for normal cellular metabolism. The best known methylating agent, a side product of cellular metabolism is SAM. Supposingly, a balance is required in the content of alkylating agents in the cell regulated by switching on/off DNA repair pathways. Authors proved that adaptive response, as it is present in *E. coli*, is not induced in *B. abortus* but the function is fulfilled by expression of *ogt* and *tag* genes regulated by *GcrA* transcription factor. Citing the work by Uphoff (PNAS, 2018) Authors conclude that also in *B. abortus* SOS is induced as the first one and "adaptive response" next to it, to repair mutations introduced by SOS. In this work there is no proof that SOS works at all in *B. abortus*.

Bellow, you'll find some specific remarks:

17 – these are in vivo not in vitro studies

26-27 – response to alkylating agents is differently regulated among bacteria e.g. *Pseudomonas putida* (vs *E. coli*)

39 – what's the source of this suggestion?

59 – these are classes of mutants

59-63 – what's the purpose of listing these mutants?

68 – I can not see the link

71 – in vivo

89 – SOS response is not LexA mediated, it is LexA-RecA regulated

96 – should be "gcrA depleted"

116 – incidentally? What does it mean?

159 – 166 – Could you specify the number of cell divisions that occurred, or measure it?

276, 305, 312 – depleted

414 – but how about SOS regulated polymerases IV and V. Again, in vivo

641 – more details. Description of bioinformatic analysis is missing

All together English needs correction.

Summing up, I think the claims of the paper are novel, however, I recommend introducing more data on the role of SOS response in *B. abortus* to complete the bacteria response to alkylation stress.

Reviewers' comments:

Reviewer #1 (Remarks to the Author):

With the exception of a few specific sections, which I will point out below, this is a well-written paper describing a very thorough and creative study that provides convincing evidence that *Brucella* strains experience alkylating stress during their intracellular residence in mammalian cells. The authors also demonstrate that these bacteria employ a mechanism unlike that of *E. coli* (which is often the paradigm of bacterial DNA repair studies) for dealing with the DNA damage caused by alkylation stress and identify the specific gene product involved in this repair. Additionally, they demonstrate that the global transcriptional regulator GcrA plays an important role in modulating the response of *Brucella* to alkylating stress. Overall, these studies provide an important contribution to our understanding of the basic biology of an important zoonotic pathogen and will have a significant impact on the *Brucella*, alpha-proteobacteria and prokaryotic DNA repair fields. But I am not convinced that they tell us a lot about the contribution of specific DNA repair pathways to *Brucella* virulence.

We thank the reviewer for his/her enthusiasm toward our paper. As he/she suggested (see below), we performed mice infection experiments, which allowed us to more specifically investigate *Brucella* virulence by showing that DNA repair pathways contribute to its virulence in this model.

The only specific concern about the present version of the manuscript is that in at least two places – e.g. lines 22-27 in the Abstract and lines 376-390 in the Discussion section, the authors should re-read their statements and revise the grammar so that the reader will understand the points that they are trying to make. Although the manuscript is in general well-written, this reviewer was confused by both of these sections.

Both sections have been rewritten. We hope they are now clearer.

The only general concern I have about the manuscript has to do with the 'virulence' studies employing the RAW264.7 cells. Although this cell line has been widely used to study the intracellular life cycle of *Brucella* strains, the replication properties in these bacteria in RAW cells do not always mimic their intracellular replication in primary explant macrophages. In fact, when one considers the link between NO, RNS and alkylation stress, it would seem more fitting to examine the intracellular replication profiles of *B. abortus* DNA repair mutants in primary explant macrophages stimulated in interferon gamma (which stimulates their iNOS activity). This reviewer would also contend that examining the spleen colonization profiles of the *B. abortus* mutants in mice would be an even more relevant way of determining the contributions of the individual gene products to virulence.

We thank the reviewer for this excellent advice, as it much improved our manuscript. We opted for an intranasal mouse infection model (Hanot Mambres et al., 2016, *J. Immunol.* **196**:3780), as this model is the closest to natural infection route, considering that Brucellosis is frequently acquired via aerosol exposure (Kaufmann et al., 1980, *Ann. N. Y. Acad. Sci.* **353**:105). In this model, *Brucella* is known to disseminate from lung cells to spleen cells (Hanot Mambres et al., 2016, *J. Immunol.* **196**:3780), which is why both organs have been studied (see new Fig 8). These data indicate that intact DNA repair pathways for alkylation damage are required for a successful infection in the lung

It was previously shown that a *B. abortus* recA mutant displays significant attenuation in the BALB/c mouse model of chronic infection (Tatum et al. 1993, *Microb. Path.* **14**:177-185), and this point needs to be mentioned in this paper and taken in consideration when the authors provide their interpretations for the data presented in Figure S8.

We were aware of those results but indeed forgot to mention them. This paper is now mentioned in the manuscript (line 340).

Reviewer #2 (Remarks to the Author):

Poncin et al. studied whether *B. abortus* experiences alkylation stress during infection and how alkylated DNA damage is repaired in this bacterium. They developed an N-nitrosation sensitive fluorescent probe and tested the survival of several mutants of *B. abortus* lacking genes involved in DNA alkylation repair. The findings are potentially interesting and novel & it is important to understand molecular mechanisms in understudied pathogens such as *B. abortus*. However, there are several issues with the work as presented in this manuscript. Additional experiments and controls will help to solidify the findings and increase the significance of this work. Specific points:

1. The N-nitrosation-sensitive probe for use in live cells is potentially a useful tool. However, crucial data and controls are missing to demonstrate the sensitivity and robustness of the method. The only data provided are in Fig. S5 and Fig. 3 showing bar plots of median fluorescence or percentage of cells with elevated fluorescence. The raw fluorescence microscopy snapshots and fluorescence distributions should be shown as well, not just the bulk values.

We thank the reviewer for pointing this out. Microscopy snapshots and distribution plots (violin plots and single dot plots) are now available for each replicate in Fig. S5.

How was the percentage of positive cells determined? Was a threshold intensity used, and if so, how was the threshold determined? Are the 23% of cells a genuine subpopulation in a bimodal distribution of fluorescence, or are these cells just the tail of a single distribution? More information has to be provided in the Materials and Methods how imaging was done for this probe.

As shown in Fig. S5b, the distribution is not convincingly bimodal. The fluorescence threshold was based on data obtained in culture (Fig S5a). It was arbitrarily set at MFI = 100, as this value is above 86% of the MFI values on labeled bacteria (in blue in Fig S5a), and below 96% of the MFI values for the positive control (labeled bacteria + KNO₂, in orange in Fig S5a). The material and method section has been improved according to this comment (lines 537-541). A more detailed protocol for MFI analysis is now available as Supporting Information S5.

Further to this point, in the discussion section it is written (Line 356): "This technique is particularly interesting because it allows the examination of the environment of each bacterial cell at a given time. Ultimately, this single-cell fluorescent approach could be further extended to other questions,..." However, no analysis on the level of single bacteria or examination of the environment was shown in this work.

All fluorescence analyses were actually performed at the single cell level, but data were presented as bar plots in order to facilitate their understanding by the reader. We apologize if this was unclear. Distribution plots have now been included in supplementary Fig S3 and S5, to highlight that data were indeed generated at the single cell level.

2. One of the key conclusions of this paper is that *B. abortus* experiences "alkylation stress" during infection. However, it is not shown that the cells actually suffer "stress". The Ada fluorescence assay and the N-nitrosation probe indicate the presence of alkylating agents but do not show that the cells are "stressed" by this. In fact, there is no change in mutagenesis during infection (Fig. S4) – alkylating agents should have left a specific mutation signature. There is also no change in CFU during infection with *B. abortus* strains that are defective in DNA alkylation repair (Fig. S8).

We understand the concerns of this reviewer about the occurrence of alkylation stress. However, it all comes to what one refers to as "stress". We believe that any change in the environment that requires a response from the bacterium can be considered as stressful. In our macrophage infection model, we do believe that *B. abortus* is reacting to alkylating stress, as the

reporter system is triggered in the early phase of infection. It is true that no mutagenesis was observable at short infection times, but it should be noted that this experiment was performed with wild type bacteria. The level of stress met in those conditions is certainly very low and fully operational repair systems were most probably enough to overcome it. As the response to alkylation stress is not linear and depends on the number and mechanisms of available repair systems (Doak et al., 2007, *Cancer Research* **67**:3904; Thomas et al., 2013, *Toxicol. Sci.* **132**:87), it is possible that the simultaneous high basal expression of several DNA repair genes in *B. abortus* (Roux et al., 2006, *J. Bacteriol.* **188**:5187) prevented the mutagenic or cytotoxic effect of those conditions. Importantly, in our mice infection experiments (see the new Fig 8 of the revised manuscript), several alkylation-specific and non-specific DNA repair pathway mutants are now reported as attenuated, indicating that in a more physiological model of infection, the stress was high enough that *B. abortus* required those functional repair systems to overcome it.

3. The authors argue that alkylation occurs specifically in the eBCV. However, this was inferred only indirectly from the time post infection. The authors need to demonstrate that the majority of *B. abortus* indeed reside in the eBCV at 5 hours PI. Especially since only 23% of cells show elevated N-nitrosation, it is important to know what fraction of cells is inside the eBCV.

In order to answer to that comment, we performed *LampI* labeling to determine what proportion of bacteria were in the eBCV (*LampI*+, as previously published [amongst others] in Comerci et al., 2001, *Cell. Microbiol.* **3**:159). The majority of *B. abortus* (about 80%) were indeed in *LampI*+ compartment (see Fig S5d). It is thus clear that even if the percentage of “positive” bacteria for nitrosation is 23%, it is likely that these bacteria are still in the eBCV compartments. Moreover, it can be seen in Fig. S5b that the distribution of signal among bacteria is not bimodal, thus the value of 23% is probably underestimated relative to the actual proportion of bacteria that experience moderate nitrosation. We chose these statistics because we wanted to be on the safe side.

4. The authors show mean fluorescence ratios of the *Ada* reporter in Fig. 2. However, given previous reports of *Ada* heterogeneous expression, it would be important to show the distribution of expression at a single cell level. Is expression limited to a small fraction of cells like the 23% of N-nitroso positive cells?

A representative experiment has now been added in Fig S3d in order to show the distribution of mean fluorescence intensities at the single cell level. It appears that in this experiment, the whole population has globally a higher level of fluorescence than in the negative control. To facilitate the compilation of the final data and their interpretation, we opted for bar plots of ratio values (Fig S3d in grey and Fig 2b).

5. The link between *GcrA* and response to alkylation stress remains quite opaque.

(i) As it is written, it is unclear if methylation by alkylation stress can affect *GcrA* activity, and thus the expression of target genes.

We have not investigated this possibility. It would be very interesting to test in future studies, but at the same time really challenging.

(ii) Depletion of *gcrA* clearly has a pleiotropic effect on cell fitness. Therefore, infection of macrophages with a *gcrA* depletion strain does not really inform whether the decrease of cell survival during infection is actually due to a deficiency in DNA repair mechanisms.

We agree with this point, as we acknowledged in lines 332-334. We still decided to include this data in our manuscript as *GcrA* is a very well conserved transcription factor in alphaproteobacteria and not much is known about its role in infection.

(iii) Fig. 5 shows that only AlkA and Tag are induced after MMS treatment (apart from the known SOS genes), but the mechanism of induction of these genes was not further investigated or discussed.

This is also a good point. After showing that there was no functional Ada-based adaptive response in *B. abortus* (Fig 4 and 5), we hypothesized that the induction of *alkA* and/or *tagA* could be regulated through the SOS system (also see point *iv* below). We thus performed new RT-qPCR experiments to compare the induction of those genes in a *lexA* overexpression background (new Fig S8). However, no statistical difference could be observed between the WT and the SOS-repressed background. Therefore, we still do not know what mechanism is responsible for those two genes induction. It would definitely be worth investigating in further studies.

(iv) Following from point iii, why did the authors not use *B. abortus* endogenous AlkA as a reporter for induction of the alkylation response, instead of using the artificial *E. coli* Ada reporter?

When we designed the experiments of this study, we did not know yet that *B. abortus alkA* was overexpressed following alkylating stress. In addition, we later suspected that this gene could be under the control of LexA, as a GTTC-N7-GTTC sequence, which is considered as a (not stringent) consensus sequence for SOS binding sites in α -proteobacteria (da Rocha et al., 2008, J. Bacteriol. **190**:1209), was present upstream of *alkA*. Moreover, the advantage of working with *E. coli ada* is that this gene and its regulation are particularly well characterized and known to be specific to alkylation stress.

6. I find the discussion section lacks focus and the writing could be much improved (I included a few suggestions in the minor points below). The authors present numerous suggestions for further experiments and speculations that are not directly supported by any experiments in this manuscript. Instead, they should concisely summarize the key findings and explain their significance.

We thank the reviewer for this comment. We tried to improve the discussion section by focusing more on the results and less on prospects.

Minor points:

1. It is surprising that *aidB* was not included in the knock out (Fig. 4) and expression (Fig. 5) experiments, since it is conserved and clearly linked to alkylation stress.

B. abortus AidB function is unclear, and its characterization was already reported in culture and in infection in a previous paper from our team (Dotreppe et al., 2011, BMC Microbiol., reference 63 in the revised manuscript and lines 325-327).

2. The alkylating agent concentrations used in vitro were much higher than those detected in vivo. For reference, can the authors show at what MMS concentration in vitro they detect similar Ada activation as in vivo, and subsequently compare cell survival at that concentration?

In order to answer to that question, we exposed *B. abortus* to low levels of MMS in culture, as suggested (see graph below). We confirmed that the response to this alkylating agent is non-linear at low doses, with the activation of the reporter system being very similar between 0.01 mM and 0.1 mM of MMS. Here, the closest dose to what we observed during macrophages infection was 0.1 mM of MMS, suggesting a very low stress as expected. Nevertheless, we suspect that the activation of the reporter system inside host cells and in culture could be different, as the environment of the eBCV is probably changing (in terms of pH, available nutrients and thus cell cycle stage, and host defense molecules). In addition, we showed that the reporter system is more reactive in exponential phase than in stationary phase (Fig S3a). We thus prefer not to include this data in the paper, as it could be confusing to the reader and subject to criticism.

3. Line 195: "The addition of ascorbate strongly decreased the mean fluorescence intensity (MFI) of labeled bacteria (Fig 3B)." This sentence probably refers to Fig. 3C not 3B. We apologize for this. It has now been modified in the revised manuscript.

4. Line 326: Usually MMR is used as abbreviation for mismatch repair. We modified the manuscript accordingly.

5. Line 362: "For example, it could occur through waves with some bacteria being affected only at a given time point." This is very speculative. What is the evidence for "waves"? We removed this line from the discussion as it was indeed too speculative.

6. Line 387: "The circle is now complete:..." This is a strange expression, and should be revised. We modified this part of the discussion.

7. Line 423: "Indeed, DNA adducts do not cause any mutation as long as replication has not occurred." The statement should be revised as it is not true as written. For example, stress induced mutagenesis occurs in stationary phase cultures. We modified the text by replacing "cause" by "fix". It is indeed true that modifications can arise on DNA in any stage of the cell cycle, but a replication event needs to occur for those modifications to cause lasting mutations.

8. Line 458: "...so it is a good indication that this bacterium probably meets the stress in the intracellular environment." The simultaneous use of "good indication" and "probably" is contradicting. This part of the manuscript has been modified accordingly.

9. Line 460: "Future research on this matter could be of great interest for the development of novel antimicrobial therapies." This statement is of little value unless additional information is provided as to what such novel therapies might involve, and how the findings in this manuscript relate to it. This sentence has been removed from the manuscript.

10. Figure 2B and 3C show the median of biological triplicates. For such small number of repeats, the mean would be a more descriptive and reliable statistic. Usually, the median is used for skewed distributions, whereas experimental repeats should follow a symmetrical distribution, i.e. best summarized by the mean. Median values of mean fluorescence values (MFI) were chosen because MFI were based on a large number of bacteria corresponding to skewed distributions (see Fig S3d and S5). By choosing median instead of mean, we wanted to minimize the bias generated by extreme MFI values.

Reviewer #3 (Remarks to the Author):

The manuscript entitled "Occurrence and repair of alkylating stress in the intracellular pathogen *Brucella abortus*" by Prof De Bolle and colleagues describes the phenomenon of the response of intracellular bacterium *Brucella abortus* to alkylating agents occurring endogenously (produced by *B. abortus*) and exogenously (inside the host cell). The work is interesting but I am not sure if Authors fully manage the subject. I have a feeling that they do not take under consideration alkylation (methylation) processes essential for normal cellular metabolism. The best known methylating agent, a side product of cellular metabolism is SAM. Supposingly, a balance is required in the content of alkylating agents in the cell regulated by switching on/off DNA repair pathways.

We thank the reviewer for raising this concern, which is fair. However, we did carefully choose what source of alkylating stress to mainly focus on, knowing that there exist several potential other sources (now discussed in the introduction section). As the reviewer states it, SAM is a well-known and well-cited natural alkylating agent, which has been shown to damage DNA in a similar way than MMS (Näslund et al., 1983, *Mutat. Res.* **119**:229) and of which the concentration has been evaluated to correspond to a continuous exposure of 20 nM of MMS in eukaryotic cells (Rydberg and Lindahl, 1982, *Embo J.* **1**:211). Another study that is a lot less cited, but very relevant to our paper, is the work by Posnick and Samson (1999, *J. Bacteriol.* **181**:6756) showing that in bacteria, a 100-fold range of SAM levels did not change the mutation rate of *E. coli*. Considering that our primary hypothesis was that host cells *actively* subject *B. abortus* to alkylating stress, we preferred to focus on the potential impact of exogenous N-nitrosation, as this source of alkylating stress is dependent on reactive nitrogen species and acidic pH (both of which could be met in the eBCV) (Iyengar et al., 1987, *Proc Natl Acad Sci U S A* **84**:6369; Ohshima et al., 1991, *Carcinogenesis* **12**:1217) and are potentially more concentrated in subcellular compartments (Espey et al., 2001, *J. Biol. Chem.* **276**:30085). Regarding the endogenous source of alkylating stress, we also chose to focus mainly on N-nitrosation because previous papers suggested that this chemical reaction was responsible for most of *E. coli* spontaneous mutations (Mackay et al., 1994, *J. Bacteriol.* **176**:3224; Taverna and Segwick, 1996, *J. Bacteriol.* **178**:5105). The experiments performed with the *B. abortus* $\Delta moaA$ strain are actually consistent with this hypothesis, even though we do not exclude that other sources of alkylating stress are at play.

Authors proved that adaptive response, as it is present in *E. coli*, is not induced in *B. abortus* but the function is fulfilled by expression of *ogt* and *tag* genes regulated by *GcrA* transcription factor. Citing the work by Uphoff (PNAS, 2018) Authors conclude that also in *B. abortus* SOS is induced as the first one and "adaptive response" next to it, to repair mutations introduced by SOS. In this work there is no proof that SOS works at all in *B. abortus*.

We thank the reviewer for this wise remark. We performed RT-qPCR experiments to strengthen our conclusion about the involvement of the SOS response (see new Fig S7 and revised manuscript, lines 251-268).

Bellow, you'll find some specific remarks:

17 – these are in vivo not in vitro studies

We modified the manuscript accordingly.

26-27 – response to alkylating agents is differently regulated among bacteria e.g. *Pseudomonas putida* (vs *E. coli*)

We agree with this statement (and added this point in the introduction, line 78). However, it remains a fact that most studies pinpoint an Ada-dependent adaptive system, with some degree of variability of the target genes and their organization in the genome, as with *P. putida* (Mielecki, 2015, *Mutat. Res. Rev. Mutat. Res.* **763**:294). Here, the originality of the paper is that we tried to decipher which other transcription factors were responsible for triggering the

response. In the case of *alkA* and *tagA*, we still do not know how they are regulated. Future studies will be necessary to answer to that question.

39 – what’s the source of this suggestion?

We did not find indication, in the literature, of mechanisms allowing immune cells to attack intracellular pathogens by alkylation. Therefore, there is no source for this suggestion.

59 – these are classes of mutants

The manuscript was modified accordingly.

59-63 – what’s the purpose of listing these mutants?

Two of them are used later in the results section and in the discussion (see Fig 3c).

68 – I can not see the link

This line has been removed from the manuscript.

71 – in vivo

The manuscript was modified accordingly.

89 – SOS response is not LexA mediated, it is LexA-RecA regulated

The manuscript was modified accordingly.

96 – should be “gcrA depleted”

“Depletion” has now been replaced by “depleted” in the manuscript.

116 – incidentally? What does it mean?

We meant “by the way”, “parenthetically”, but since this word is apparently confusing, we removed it from the manuscript.

159 – 166 – Could you specify the number of cell divisions that occurred, or measure it?

We can roughly estimate the number of cell divisions at 7 to 8 at 48 h post infection (PI) in macrophages and in culture. Indeed, in infection, from 5 h PI (non-dividing stage) to 48 h PI, we observe an increase of CFU of about 2 log (see Fig S9 for e.g.), corresponding to about 7 cell divisions, if there is no bacterial death. In liquid culture, *B. abortus* typically goes from OD_{600nm} 0.1 to 1.6 (4 cell divisions) in 24 h (data not shown). As the sequenced colonies were picked after two dilutions in 48 h, we can estimate that they underwent about 8 cell divisions, again assuming that no bacterial deaths occurred. The time 60 h for mice infection was picked because we knew that the infection process is slightly delayed *in vivo* compared to *in vitro* (E. Muraille’s personal communication, unpublished). Since these estimates, although reasonable and likely, are not precise because the proportion of bacterial death is unknown, we decided to not include them in the revised manuscript.

276, 305, 312 – depleted

“Depletion” has now been replaced by “depleted” in the manuscript.

414 – but how about SOS regulated polymerases IV and V. Again, in vivo

We have included *dinB* (pol IV) and *imuABC* (functional equivalent of pol V in alpha-proteobacteria) in our RT-qPCR experiments (Fig S7).

641 – more details. Description of bioinformatic analysis is missing

More information is now available in the material and methods.

All together English needs correction.

We tried our best to provide a revised manuscript with better English.

Summing up, I think the claims of the paper are novel, however, I recommend introducing more data on the role of SOS response in *B. abortus* to complete the bacteria response to alkylation stress.

We thank the reviewer for his/her suggestion on investigating the SOS response further, as it helped us to improve the manuscript.

REVIEWERS' COMMENTS:

Reviewer #1 (Remarks to the Author):

My concerns have been addressed.

Reviewer #2 (Remarks to the Author):

The revised manuscript addresses most of my previous comments, and I also appreciate the inclusion of the mouse infection experiments in response to the other reviewer's concerns. The role of GcrA in regulating expression of alkylation repair genes still remains unclear to me, but I agree that the data are nevertheless interesting for the field and should be published. The writing is also improved.

Reviewer #3 (Remarks to the Author):

I fully accept the revisions made by the authors

Response to reviewers

=> We would like to thank the three reviewers once more about their comments, as they greatly helped us to improve our paper.

REVIEWERS' COMMENTS:

Reviewer #1 (Remarks to the Author):

My concerns have been addressed.

Reviewer #2 (Remarks to the Author):

The revised manuscript addresses most of my previous comments, and I also appreciate the inclusion of the mouse infection experiments in response to the other reviewer's concerns. The role of GcrA in regulating expression of alkylation repair genes still remains unclear to me, but I agree that the data are nevertheless interesting for the field and should be published. The writing is also improved.

Reviewer #3 (Remarks to the Author):

I fully accept the revisions made by the authors